# Deep evolutionary origin of gamete-directed zygote activation by KNOX/BELL transcription factors in green plants

Tetsuya Hisanaga[1,2], Shota Fujimoto[1], Yihui Cui[1], Katsutoshi Sato[1], Ryosuke Sano[1], Shohei Yamaoka[3], Takayuki Kohchi[3], Frédéric Berger[2], Keiji Nakajima[1]*

[1]Graduate School of Science and Technology, Nara Institute of Science and Technology, Nara, Japan; [2]Gregor Mendel Institute (GMI), Austrian Academy of Sciences, Vienna Biocenter, Vienna, Austria; [3]Graduate School of Biostudies, Kyoto University, Kyoto, Japan

**Abstract** KNOX and BELL transcription factors regulate distinct steps of diploid development in plants. In the green alga *Chlamydomonas reinhardtii,* KNOX and BELL proteins are inherited by gametes of the opposite mating types and heterodimerize in zygotes to activate diploid development. By contrast, in land plants such as *Physcomitrium patens* and *Arabidopsis thaliana*, KNOX and BELL proteins function in sporophyte and spore formation, meristem maintenance and organogenesis during the later stages of diploid development. However, whether the contrasting functions of KNOX and BELL were acquired independently in algae and land plants is currently unknown. Here, we show that in the basal land plant species *Marchantia polymorpha*, gamete-expressed *KNOX* and *BELL* are required to initiate zygotic development by promoting nuclear fusion in a manner strikingly similar to that in *C. reinhardtii*. Our results indicate that zygote activation is the ancestral role of KNOX/BELL transcription factors, which shifted toward meristem maintenance as land plants evolved.

*For correspondence:
k-nakaji@bs.naist.jp

**Competing interest:** The authors declare that no competing interests exist.

## Introduction

The life cycles of eukaryotes alternate between diploid (2n ) and haploid (n) phases through meiosis and fertilization (*Bowman et al., 2016*). In land plants, both the haploid and diploid phases are multicellular, producing gametophytic and sporophytic bodies, respectively. In bryophytes including liverworts, mosses, and hornworts, gametophytes are larger and more morphologically complex than sporophytes, which consist of only a few cell types. During the course of land plant evolution, the life cycle shifted toward a sporophyte-dominant style, presumably to facilitate adaptation to terrestrial environments where it is advantageous to generate large sporophytes that produce many spores. Consequently, the sporophytes of extant flowering plants (angiosperms) exhibit far more complex morphologies than their male and female gametophytes, the pollen grain, and embryo sac, respectively, which are composed of only a few cells. This evolutionary transition in life cycle is thought to have been facilitated by the cooption of genes and/or gene regulatory networks that regulate gametophyte development to function in sporophyte development (*Bowman et al., 2019*). However, key steps of life cycle progression per se have continued to be driven by conserved regulators during land plant evolution, as recently reported for gametophytic sexual differentiation and gamete formation (*Koi et al., 2016*; *Rövekamp et al., 2016*, *Higo et al., 2018*; *Yamaoka et al., 2018*; *Hisanaga et al., 2019a*; *Hisanaga et al., 2019b*).

Homeodomain transcription factors (HD TFs) are developmental regulators that are evolutionarily conserved in eukaryotes. HD TFs are classified into two families, three-amino-acid-loop-extension

(TALE) and non-TALE, based on amino acid sequence similarity in the HD domain (*Bertolino et al.,* *1995*; *Derelle et al., 2007*). In the fungi *Saccharomyces cerevisiae* and *Coprinopsis cinerea*, TALE and non-TALE HD TFs are expressed in haploid cells of opposite mating types (*Herskowitz, 1989*; *Kues* *et al., 1992*). These proteins heterodimerize in zygotes to regulate the expression of genes promoting the haloid-to-diploid transition (*Goutte and Johnson, 1988*; *Spit et al., 1998*). In green plants, TALE HD TFs have diversified into the KNOX (KNOTTED1-LIKE HOMEOBOX) and BELL (BELL-LIKE) subfamilies. In the unicellular green alga *Chlamydomonas reinhardtii*, the KNOX protein GAMETE SPECIFIC MINUS1 (GSM1) and the BELL protein GAMETE SPECIFIC PLUS1 (GSP1) accumulate in the cytosol of *minus* and *plus* gametes, respectively. Upon fertilization, the two proteins heterodimerize and translocate to both male and female pronuclei to activate the expression of early zygote-specific genes. Loss-of-function mutations in either *GSP1* or *GSM1* result in pleiotropic phenotypes involving cellular rearrangements in zygotes, such as the loss of nuclear and mitochondrial fusion, lack of selective degradation of *minus*-derived chloroplast DNA and chloroplast membrane fusion, and defects in flagellar resorption (*Joo et al., 2017*; *Kariyawasam et al., 2019*; *Lee et al., 2008*; *Lopez et al., 2015*; *Nishimura et al., 2012*).

In land plants, KNOX proteins are further diversified into class I (KNOX1) and class II (KNOX2) (*Kerstetter et al., 1994*; *Mukherjee et al., 2009*). The developmental functions of these proteins have been studied extensively in angiosperms such as maize (*Zea mays*), rice (*Oryza sativa*), and *Arabidopsis* (*Arabidopsis thaliana*) (reviewed in *Hay and Tsiantis, 2010*). Based on their expression patterns and the phenotypes of both knockout and overexpression lines, KNOX1 proteins are thought to promote cell proliferation in the meristematic tissues of aerial organs. The biological functions of *KNOX2* genes are somewhat elusive, but they are thought to act antagonistically to *KNOX1* to promote cell differentiation (*Furumizu et al., 2015*).

The apparent functional dissimilarity of KNOX proteins between *C. reinhardtii* and *Arabidopsis* (zygote activation versus cellular proliferation/differentiation) may reflect the large phylogenetic distance between these two species as they separated into two major green plant lineages, Chlorophyta and Streptophyta, some 700 million years ago (*Becker, 2013*). Functional analyses of *KNOX* genes in the moss *Physcomitrium patens,* however, pointed to some commonality between the two *KNOX* functions in sporophyte development. The moss genome contains three *KNOX1* and two *KNOX2* genes, which are all primarily expressed in sporophytes, though expression of one *KNOX1* and two *KNOX2* genes is additionally detected in egg cells (*Horst et al., 2016*; *Sakakibara et al.,* *2013*; *Sakakibara et al., 2008*). A triple loss-of-function mutant of all three *KNOX1* genes was defective in cell division and differentiation in sporophytes, as well as spore formation (*Sakakibara et al.,* *2008*). By contrast, simultaneous knockout of the two *KNOX2* genes resulted in ectopic gametophyte formation in sporophyte bodies (*Sakakibara et al., 2013*). Thus, at least in one bryophyte species, KNOX1 and KNOX2 control sporophyte development via two pathways, with one ensuring proper sporophyte development (like *C. reinhardtii* GSM1) and the other promoting cell proliferation (like *Arabidopsis* KNOX1 proteins). While the transition of the role of KNOX/BELL from zygote activation to sporophyte morphogenesis likely arose during plant evolution, the point in plant phylogeny at which this transition occurred is unclear.

Here, we analyzed the roles of KNOX1 and BELL in the liverwort *Marchantia polymorpha*, a model species suitable to study evolution of sexual reproduction in plants (*Hisanaga et al., 2019b*). We uncovered unexpected conservation of KNOX/BELL function between the phylogenetically distant green plants *M. polymorpha* and *C. reinhardtii*, but not between the more closely related *M. poly-* *morpha* and *P. patens*. Thus, the functional transition of KNOX/BELL from zygote activation to sporophyte morphogenesis occurred at least once in the land plant lineage independently of the acquisition of multicellular sporophytes. Additionally, we uncovered inverted sex-specific expression patterns of *KNOX* and *BELL* genes between *C. reinhardtii* and *M. polymorpha*, suggesting that anisogamy evolved independently of *KNOX/BELL* expression in gametes.

## Results

### Mp*KNOX1* is an egg-specific gene in *M. polymorpha*

We previously reported that an RWP-RK TF MpRKD promotes egg cell differentiation in *M. poly-* *morpha*. Loss-of-function Mp*rkd* mutant females grow normally and produce archegonia like the

wild-type, but their egg cells do not mature, instead degenerating after ectopic cell division and vacuolization (**Koi et al., 2016**). We made use of this egg-specific defect in Mp*rkd* to identify genes preferentially expressed in egg cells of *M. polymorpha*. Briefly, we collected ~2000 archegonia from two independent Mp*rkd* female mutant lines (Mp*rkd-1* and Mp*rkd-3*; **Koi et al., 2016**), each in two replicates. As a control, ~4000 archegonia were collected from wild-type females in four replicates. We extracted RNA from each pool and analyzed it by next-generation sequencing. Comparative transcriptome analysis identified 1583 and 170 genes with significantly decreased and increased mRNA levels, respectively, in Mp*rkd* compared to wild-type archegonia, respectively (≥3 -fold and *false discovery rate* <0.01; **Figure 1A** and **Figure 1—figure supplement 1A**, **Supplementary file 1**). Among the genes with reduced expression levels, Mp*KNOX1* (Mp5g01600), the only class I *KNOX* gene in *M. polymorpha* (**Bowman et al., 2017**, **Frangedakis et al., 2017**), showed more than 300-fold reduced expression level in Mp*rkd* as compared to that in the wild-type (fragments per kilobase of exon per million mapped reads (FPKM) values are 19.5, 0.0629, and 0.0124 in wild-type, Mp*rkd-1*, and Mp*rkd-3* archegonia, respectively; **Supplementary file 1**). The MpKNOX1 polypeptide contains KNOX I, KNOX II, ELK, and Homeobox domains, as do KNOX proteins from green algae, mosses, ferns, and flowering plants (**Figure 1B**).

Previous RNA-sequencing data (**Bowman et al., 2017**) indicated that Mp*KNOX1* is specifically expressed in female plants (**Figure 1—figure supplement 1B**). To obtain the detailed expression patterns of Mp*KNOX1*, we performed RT-PCR analysis using RNA extracted from vegetative and reproductive organs of male and female gametophytes, as well as 3-week-old sporophytes, and confirmed the specific expression of Mp*KNOX1* in archegoniophores (**Figure 2A**). No expression was detected in female or male thalli (leaf-like vegetative organs), antheridiophores (male reproductive branches), or sporophytes (**Figure 2A**). To visualize the cell- and tissue-specific expression patterns of Mp*KNOX1*, we generated a Mp*KNOX1* transcriptional reporter line (*MpKNOX1pro:H2B-GFP*). Consistent with the >300 -fold reduced Mp*KNOX1* transcript levels in egg-deficient Mp*rkd* archegonia (**Figure 1B**), reporter expression was specifically detected in egg cells (**Figure 2C**) and not in developing archegonia before the formation of egg progenitors (**Figure 2B**). Together, these data indicate that Mp*KNOX1* is an egg-specific gene in *M. polymorpha*.

## Zygote activation of wild-type *M. polymorpha* lags during karyogamy

The egg-specific expression of Mp*KNOX1* attracted our attention as this pattern is in contrast to the previously reported function of KNOX1s in sporophyte morphogenesis in land plants (**Furumizu et al., 2015**; **Sakakibara et al., 2008**). Instead, the egg-specific expression of Mp*KNOX1* is reminiscent of *GSM1*, a KNOX homolog in *C. reinhardtii* that is expressed in *plus* gametes and activates zygote development after fertilization (**Joo et al., 2017**; **Kariyawasam et al., 2019**; **Lee et al., 2008**; **Lopez et al., 2015**; **Ning et al., 2013**; **Nishimura et al., 2012**). Therefore, we explored whether Mp*KNOX1* plays a role in zygote activation in *M. polymorpha*.

While the processes of gametogenesis and embryo patterning in *M. polymorpha* (**Figure 3A**) have been characterized histologically (**Durand, 1908**; **Higo et al., 2016**; **Koi et al., 2016**; **Shimamura, 2016**; **Zinsmeister and Carothers, 1974**), the subcellular dynamics associated with zygote activation have not been described in detail. To visualize these dynamics, we established a simple in vitro fertilization method for *M. polymorpha*. Briefly, several antheridiophores and archegoniophores were co-cultured for 1 hr in a plastic tube containing an aliquot of water to allow sperm to be released from the antheridia and enter into the archegonia (**Figure 3B**). At the end of the co-culture period, sperm nuclei were visible in most eggs (**Figure 3C**), indicating that fertilization had been completed during the 1 hr of co-culturing. Subsequently, sperm-containing archegonia were transferred to a fresh tube containing water and cultured for 2 weeks (**Figure 3B**). This experimental regime restricted the timing of sperm entry to the 1 hr time window of the co-culture period (whose termination is defined hereafter as the time of fertilization), allowing us to perform a time-course observation of fertilization and embryogenesis (**Figure 3D–I**).

We observed the cellular dynamics of zygotes and early embryos using optimized cell wall staining and tissue clearing techniques (**Miyashima et al., 2019**; **Kurihara et al., 2015**). No cell walls were stained in mature egg cells (**Figure 3—figure supplement 1A**), indicating that cell walls are not present in mature eggs, a prerequisite for fusion with sperm cells. A cell wall around the zygote was detected only at 3 days after fertilization (DAF) (**Figure 3—figure supplement 1B–D**), whereas the first

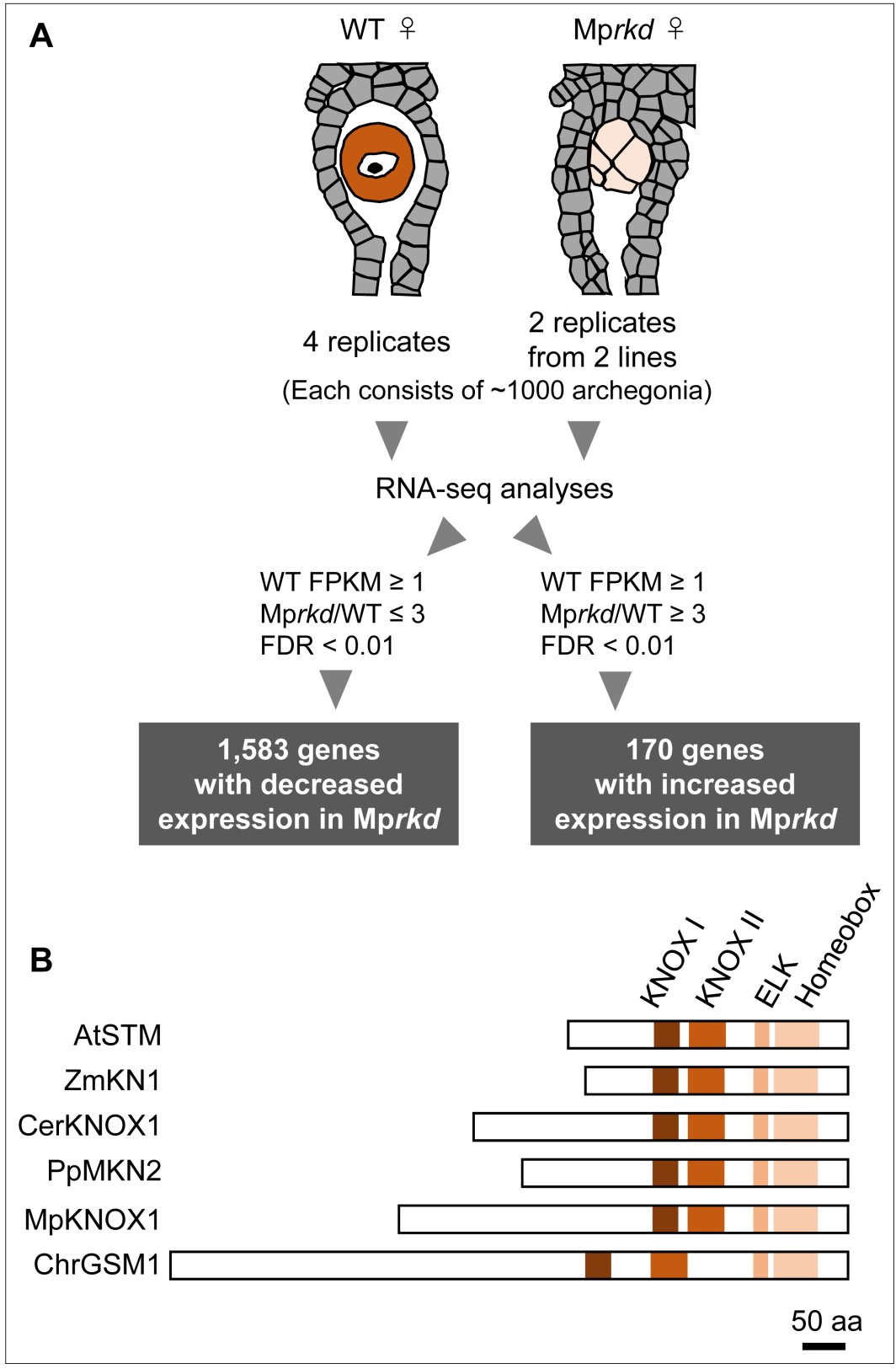

**Figure 1.** Comparative transcriptome analysis of wild-type and Mp*rkd* archegonia and identification of Mp*KNOX1* as an egg-specific gene. (**A**) Schematic illustration of RNA-seq analysis comparing the archegonia transcriptomes from wild-type females and egg-deficient Mp*rkd* mutant females. About 4000 and 2000 archegonia were collected from wild-type and each of the two Mp*rkd* mutant lines, and randomly allocated into four and two

*Figure 1 continued on next page*

*Figure 1 continued*

replicates, respectively. (**B**) Comparison of the domain arrangements of MpKNOX1 vs. representative class I KNOX proteins from *Arabidopsis thaliana* (AtSTM; GenBank accession number AEE33958.1), *Zea mays* (ZmKN1; AAP21616.1), *Ceratopteris richardii* (CerKNOX1, BAB18582.1), *Physcomitrium patens* (PpMKN2; AAK61308.2), and *Chlamydomonas reinhardtii* (ChrGSM; ABJ15867.1).

The online version of this article includes the following figure supplement(s) for figure 1:

**Figure supplement 1.** Comparative transcriptome analysis of wild-type and Mp*rkd* archegonia and identification of Mp*KNOX1* as an egg-specific gene in *M. polymorpha*.

zygotic division occurred at 4–5 DAF (***Figure 3—figure supplement 1E and F***). At 1–3 DAF, a male pronucleus was clearly stained with DAPI, in contrast to the female pronucleus (not visible by DAPI staining), and was typically positioned halfway between the periphery and center of the fertilized egg (***Figure 3D and E***, green arrowhead). At 4 DAF, most zygotes completed the first division (***Figure 3F***), indicating that karyogamy takes place at 3–4 DAF. After 5 DAF, zygotes and the surrounding archegonial wall cells divided to form sporophytes and the calyptra (a protective gametophyte tissue), respectively (***Figure 3G–I***). These cellular rearrangements, including karyogamy and embryogenesis, proceeded at a rate comparable to that in zygotes produced in planta (***Figure 3—figure supplement 2***), confirming that our in vitro fertilization protocol faithfully recapitulated fertilization programs in planta.

## Maternal Mp*KNOX1* is required for pronuclear fusion in zygotes

To analyze the biological functions of Mp*KNOX1*, we generated loss-of-function mutants of Mp*KNOX1* using a CRISPR/Cas9 technique optimized for *M. polymorpha* (***Sugano et al., 2018***). We obtained three independent female mutant lines that harbored nucleotide insertions or deletions resulting in premature stop codons preceding the region encoding the HD (***Figure 4—figure supplement 1A***). All mutants were indistinguishable from wild-type females in terms of both vegetative and reproductive morphology (***Figure 4—figure supplement 2***). Mature archegonia and egg cells of the Mp*knox1* mutants were also indistinguishable from those of the wild-type (***Figure 4A and E***), indicating that Mp*KNOX1* functions are dispensable for both gametophyte development and gametogenesis.

We crossed the Mp*knox1* mutant females with wild-type males and observed the resulting zygotes by microcopy. Similar to wild-type zygotes, each Mp*knox1* egg fertilized with wild-type sperm harbored a male pronucleus at 1 DAF (***Figure 4B, F and I***), indicating that Mp*knox1* eggs are able to fuse with wild-type sperm and support decondensation of sperm nuclei. At 5 DAF, however, male and female pronuclei remained unfused in fertilized Mp*knox1* eggs (100%, n = 34–38, ***Figure 4G and J***), in contrast with wild-type fertilized eggs, a majority of which were undergoing sporophyte development (89%, n = 38, ***Figure 4C and J***). At 7 DAF, most Mp*knox1* eggs contained unfused male and female pronuclei (87–94%, n = 31–56, ***Figure 4H and K***). The three independent Mp*knox1* mutant lines exhibited indistinguishable defects in karyogamy and sporophyte development (***Figure 4I–K***). Importantly, these defects were rescued in archegonia expressing MpKNOX1-GFP by the Mp*KNOX1* promoter (*gMpKNOX1-GFP*, ***Figure 4—figure supplement 3***), confirming the notion that Mp*KNOX1* is required maternally to complete fertilization.

A small fraction of Mp*knox1* eggs fertilized with wild-type sperm developed into sporangia that produced functional spores (***Figure 4—figure supplement 4***), suggesting that redundant genetic pathway(s) can compensate for the loss of Mp*KNOX1* and/or that our Mp*knox1* mutant alleles were not null, though genomic sequences preceding the HD-coding region were disrupted. This residual embryogenic capacity allowed us to obtain male Mp*knox1* gametophytes from the resulting spores. The male Mp*knox1* mutants produced functional sperm capable of producing normal embryos when used to fertilize wild-type eggs (***Figure 4—figure supplement 5***), indicating that paternal Mp*KNOX1* is dispensable for gametophyte development, fertilization, and embryogenesis. To further examine the lack of paternal contribution of Mp*KNOX1* to fertilization and sporophyte development, we crossed Mp*knox1* females with Mp*knox1* males. Resulting homozygous Mp*knox1* zygotes exhibited the karyogamy and sporangia formation defects indistinguishable from those observed in zygotes carrying maternally inherited Mp*knox1* alone (***Figure 4—figure supplements 4 and 6***). Together, these results indicate that the egg-derived functional Mp*KNOX1* allele or its protein products, but not

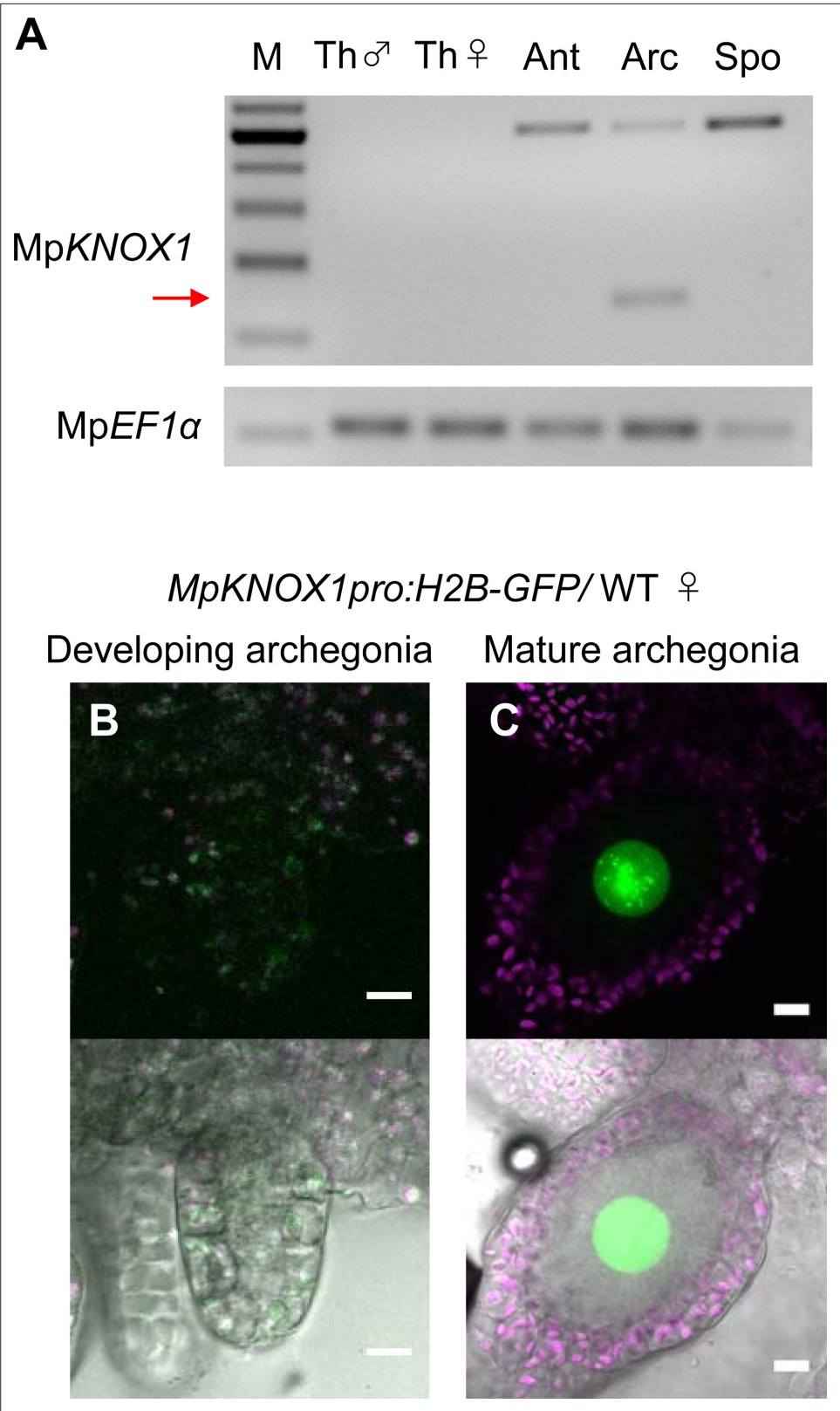

**Figure 2.** Mp*KNOX1* is specifically expressed in egg cells. (**A**) RT-PCR analysis of Mp*KNOX1*. Lanes are labeled as follows: M: size markers; Th ♂: male thalli; Th ♀: female thalli; Ant: antheridiophores; Arc: archegoniophores; Spo: sporophytes of 3-week-old plants. Constitutively expressed Mp*EF1α* was used as a control. Red arrow indicates the expected size of PCR products from spliced Mp*KNOX1* mRNA. Bands at the top of the gel likely correspond to

*Figure 2 continued on next page*

*Figure 2 continued*

unspliced Mp*KNOX1* transcripts. Shown is a representative result from the experiments using three independently collected plant samples each with two technical replicates (two PCRs from each cDNA pool). See *Figure 4—figure supplement 1A* for the primer position. (**B, C**) Expression of the Mp*KNOX1* transcriptional reporter. Magenta: chlorophyll autofluorescence; green: GFP fluorescence. Lower panels are merged photographs of fluorescence and bright-field images. Bars, 10 µm.

those derived from sperm, are required to activate zygote development, more specifically karyogamy, during *M. polymorpha* fertilization.

In both animals and plants, karyogamy occurs via a two-step process: pronuclear migration and nuclear membrane fusion (*Fatema et al., 2019*). During pronuclear migration, one or both pronuclei migrate to become in close proximity, while during nuclear membrane fusion, the nuclear envelopes of the two pronuclei fuse together to produce a zygotic nucleus with both maternal and paternal genomes. To identify which of these steps is affected by the Mp*knox1* mutation, we visualized the pronuclear envelope by expressing GFP-tagged Sad1/UNC84 (SUN) domain-containing proteins (MpSUN; Mp5g02400) under the control of an egg-specific promoter (*ECpro:MpSUN-GFP*, see Materials and methods for details). The SUN proteins are known to localize specifically to nuclear envelope in animals, yeasts, and vascular plants (*Graumann et al., 2010*; *Tzur et al., 2006*). Before fertilization, wild-type and Mp*knox1* egg nuclei were of a similar size (approximately 20 µm in diameter) and were surrounded by a mesh-like membranous structure (*Figure 4L and O*). At 3 DAF, *ECpro:MpSUN-GFP* signals were visualized at the surfaces of both female and male pronuclei, suggesting the presence of a nuclear envelope (*Figure 4M and P*). In both wild-type- and Mp*knox1*-derived zygotes, female and male pronuclei were tethered to each other by a membranous structure marked by MpSUN-GFP (*Figure 4M and P*). At 5 DAF, however, male and female pronuclei remained separated by an intercalating membranous structure in Mp*knox1*-derived zygotes, while those from wild-type eggs already showed two embryonic nuclei (*Figure 4N and Q*). Together, these observations indicate that maternal Mp*KNOX1* or its protein product is dispensable for the organization and migration of pronuclei but is required for pronuclear membrane fusion.

## Both maternal and paternal Mp*BELL* alleles contribute to karyogamy

KNOX proteins heterodimerize with BELL proteins to regulate gene transcription (*Hay and Tsiantis, 2010*). In *C. reinhardtii*, a *minus* gamete-derived KNOX protein (GSM1) heterodimerizes with a *plus* gamete-derived BELL protein (GSP1) upon fertilization and activates the majority of early zygote-specific genes (*Joo et al., 2017*; *Kariyawasam et al., 2019*; *Lee et al., 2008*; *Lopez et al., 2015*; *Nishimura et al., 2012*). Loss of GSM1 and/or GSP1 results in pronuclear fusion arrest, a phenotype similar to that of Mp*knox1* mutants. The phenotypic similarity between *M. polymorpha* Mp*knox1* and *C. reinhardtii gsm1* mutants suggests that the role of KNOX and BELL in zygote activation is conserved and that its evolutionary origin can be traced back to a common ancestor of the two species.

In support of this hypothesis, publicly available transcriptome data (*Bowman et al., 2017*) indicate that two of the five *BELL* genes of *M. polymorpha*, Mp*BELL3* (Mp8g02970) and Mp*BELL4* (Mp8g07680), are preferentially expressed in the antheridiophores of male plants, whereas Mp*BELL1* (Mp8g18310) and Mp*BELL5* (Mp5g11060) are preferentially expressed in sporophytes and archegonia, respectively (*Figure 5—figure supplement 1*). RT-PCR analysis confirmed that Mp*BELL3* and Mp*BELL4* are specifically expressed in antheridiophores containing antheridia (*Figure 5A*).

To investigate whether Mp*BELL3* and/or Mp*BELL4* are required for fertilization, we generated two male and one female mutant lines in which Mp*BELL3* and Mp*BELL4* were simultaneously disrupted (hereafter referred to as Mp*bell3/4*) using a CRISPR/Cas9 nickase system (*Figure 4— figure supplement 1B and C*). Both female and male Mp*bell3/4* gametophytes grew normally (*Figure 4—figure supplement 2*) and produced normal gametangia and gametes capable of fertilization (*Figure 5—figure supplement 2*, *Figure 5B, D, G, J and L*). By contrast, approximately a half of zygotes produced by crossing wild-type females and Mp*bell3/4* males did not complete

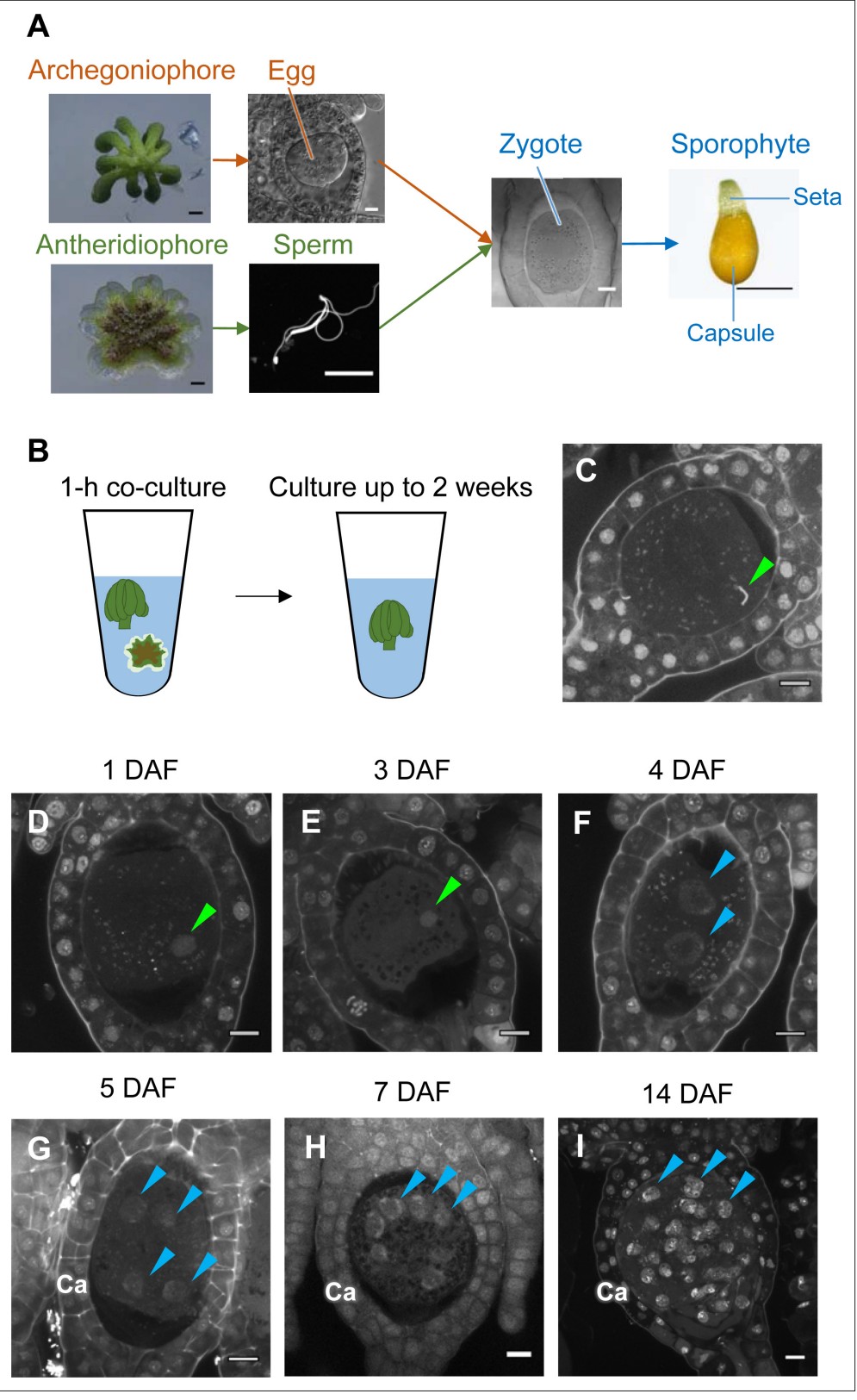

**Figure 3.** Time-course observation of subcellular dynamics during *M. polymorpha* fertilization. (**A**) Schematic representation of sexual reproduction in *M. polymorpha*. Female plants develop umbrella-shaped sexual branches termed archegoniophores that form egg-containing archegonia. Male plants develop disc-shaped sexual branches termed antheridiophores that form antheridia, which produce numerous motile sperm cells.

*Figure 3 continued on next page*

*Figure 3 continued*

Upon soaking antheridiophores in water, the sperm cells are released from the antheridia and swim to egg cells in the archegonia. After fertilization, each zygote undergoes embryogenesis by dividing and differentiating into a sporophyte body consisting of a capsule containing haploid spores and a short supportive stalk called the seta. Black bars, 1 mm. White bars, 10 μm. (**B**) Illustration of the in vitro fertilization method used in this study. Excised archegoniophores and antheridiophores were co-cultured in water for 1 hr to allow fertilization to take place. The archegoniophores were transferred to a fresh tube containing water for further culturing. The tube lids were left open to allow gas exchange to occur. Archegoniophores containing sporophytes were cultured for up to 2 weeks. (**C**) A DAPI-stained zygote after 1 hr of co-culture. Most zygotes contained sperm nuclei at this time (green arrowhead). Bars, 10 μm. (**D–I**) DAPI-stained zygotes and sporophytes at the indicated days after fertilization (DAF). Male pronuclei (green arrowheads) were visible at 1–3 DAF (**D, E**) in wild-type fertilized eggs. In most zygotes, karyogamy was completed, and cells were cleaved at 4 DAF (**F**). Sporophyte cells continued to divide at 5–14 DAF (**G–I**), as visualized by the presence of multiple nuclei (blue arrowheads; not all nuclei are labeled in **H** and **I**). Ca: calyptra. Bars, 10 μm.

The online version of this article includes the following figure supplement(s) for figure 3:

**Figure supplement 1.** Cell wall regeneration during zygote development.

**Figure supplement 2.** Cellular dynamics of zygotes and embryos generated by in planta crossing.

---

karyogamy by 5 DAF (55% and 37% for Mp*bell3/4-1^ge* and Mp*bell3/4-2^ge*, respectively; *Figure 5E, F and M*).

Somewhat unexpectedly, zygotes obtained from a reciprocal cross (Mp*bell3/4* females with wild-type males) exhibited a low but significant degree of karyogamy arrest (30% and 22% for Mp*bell3/4-1^ge* andMp*bell3/4-2^ge*, respectively; *Figure 5H, I and M*), despite the lack of detectable Mp*BELL3/4* expression in female gametes. Furthermore, a majority of 5-DAF zygotes obtained from a cross between Mp*bell3/4* females and Mp*bell3/4* males were arrested at karyogamy (75% and 92% for Mp*bell3/4-1^ge* and Mp*bell3/4-2^ge*, respectively; *Figure 5K and M*). A similar but slightly less proportion of arrested zygotes was observed at 7 DAF for each genotype (*Figure 5—figure supplement 3A–G*). Moreover, unfused pronuclei were not observed in any of the developing embryos harboring Mp*bell3/4* allele(s) (*Figure 5—figure supplement 3C and E*). Consistently, the number of sporangia in Mp*bell3/4* homozygotes was about a third of that in wild-type (*Figure 5—figure supplement 3H*). Taken together, these results suggest that not only paternally inherited Mp*BELL3* and/or Mp*BELL4* alleles(s) and/or their protein products, but also maternally inherited Mp*BELL3* and/or Mp*BELL4* allele(s) contribute to sporophyte development. A correlation between the proportion of arrested zygotes and the degree of reduction in sporangia number in each genotype suggests that MpBELL3/4 mainly act to promote karyogamy for the progression of sporophyte development.

## MpKNOX1 proteins transiently localize to male and female pronuclei in an Mp*BELL3/4*-dependent manner

Our genetic analyses indicated that Mp*KNOX1* functions after fertilization despite its specific transcription in unfertilized egg cells. This observation suggests that MpKNOX1 protein produced in unfertilized egg cells functions in zygotes. To test this hypothesis, we analyzed MpKNOX1 protein dynamics during fertilization by examining *gMpKNOX1-GFP* plants by confocal microscopy. In mature egg cells, MpKNOX1-GFP was specifically detected in the cytosol (*Figure 6A*). After crossing with a wild-type male, GFP signals were detected in both male and female pronuclei at 12 hours after fertilization (HAF) (83%, n = 48, *Figure 6B and G*). At 24 HAF, the GFP signals were totally excluded from the pronuclei (*Figure 6C*). These observations suggest that upon fertilization MpKNOX1 translocates from the cytosol to pronuclei.

As KNOX proteins function as transcription factors, the transient localization of MpKNOX1 in pronuclei should be critical for its role in regulating karyogamy-promoting genes. In *Arabidopsis* and Chlamydomonas, KNOX proteins are recruited to nuclei through interactions with BELL proteins. To examine whether paternally inherited Mp*BELL3* and/or Mp*BELL4* contribute to the pronuclear localization of MpKNOX1, we crossed *gMpKNOX1-GFP* females with wild-type or Mp*bell3/4* males and analyzed the subcellular localization of MpKNOX1-GFP in zygotes. In contrast to zygotes derived from a cross with wild-type males, where GFP signals were preferentially detected in male

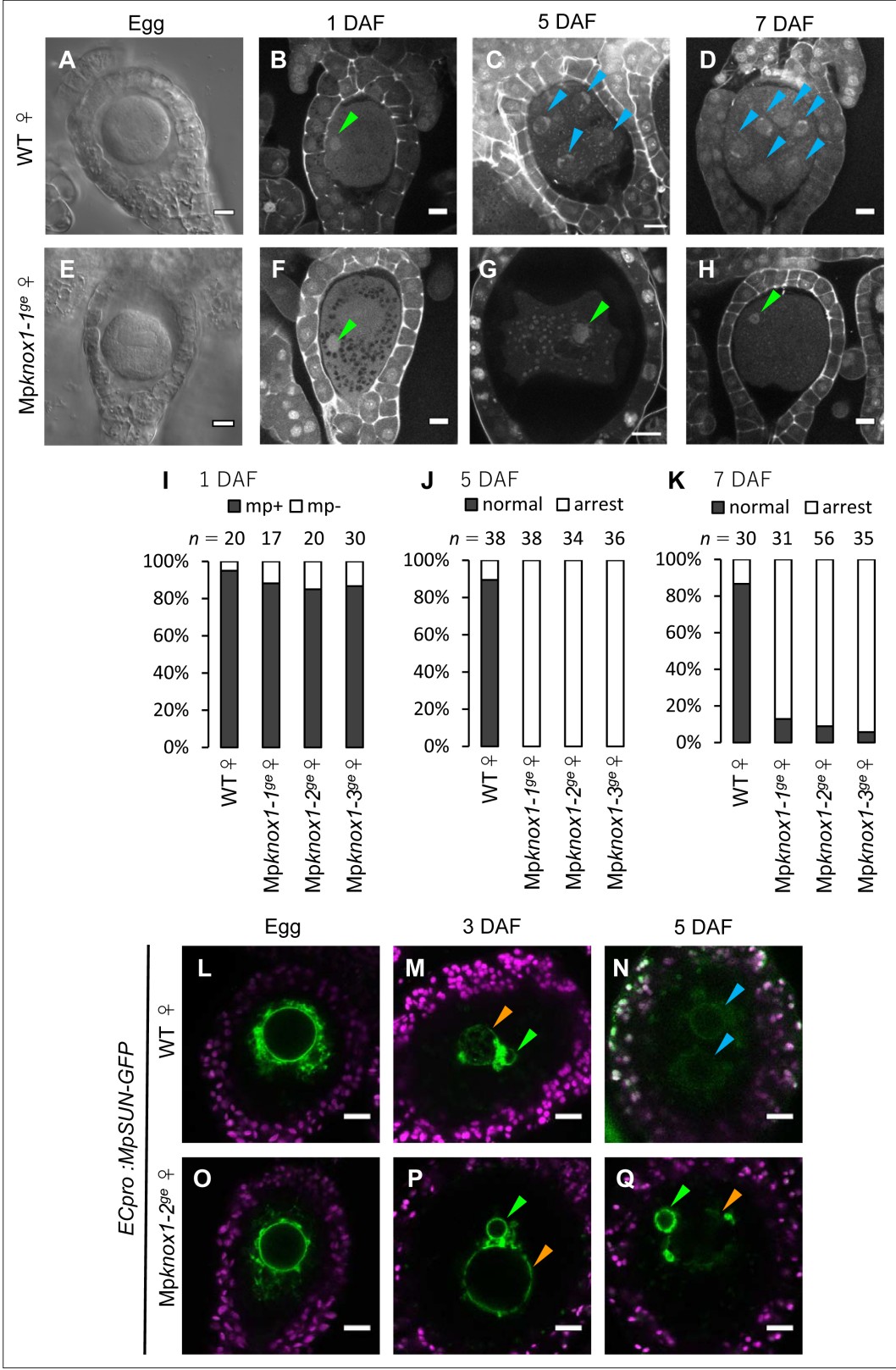

**Figure 4.** Maternally inherited Mp*KNOX1* is required for nuclear fusion. (**A, E**) Bright-field images of wild-type (WT; **A**) and Mp*knox1-1^{ge}* (**E**) archegonia. (**B–D and F–H**). 1 day after fertilization (DAF) (**B, F**), 5 DAF (**C, G**), and 7 DAF (**D, H**) zygotes from a cross between WT female and male plants (**B–D**), and a cross between Mp*knox1-1^{ge}* female and WT male plants (**F–H**), indicating that maternal Mp*KNOX1* is dispensable for fertilization (**B, F**) but is required

*Figure 4 continued on next page*

*Figure 4 continued*

for embryogenesis (**C, D, G, H**). (**I–K**) Bar graphs showing the ratios of zygotes containing male pronuclei (mp+) vs. those not containing male pronuclei (mp-) at 1 DAF (**I**), and developed vs. arrested zygotes at 5 DAF (**J**) and 7 DAF (**K**) derived from the crosses of either WT or Mp*knox1* females with WT males. Number of observed zygotes is shown above each bar. (**L–Q**) Egg cells of *MpSUN-GFP* marker lines in the WT (**L**) or Mp*knox1-2^{ge}* (**O**) female background were crossed with WT males. At 3 DAF, male and female pronuclei were in contact with each other in both WT (**M**) and Mp*knox1* (**P**) eggs. At 5 DAF, zygotes derived from a WT egg started to divide (**N**), while those from an Mp*knox1* female (**Q**) were arrested without nuclear membrane fusion. Green arrowhead: male pronucleus; orange arrowhead: female pronucleus; blue arrowhead: embryo nucleus. Bars, 10 μm.

The online version of this article includes the following figure supplement(s) for figure 4:

**Figure supplement 1.** Generation of loss-of-function mutant lines by CRISPR/Cas9.

**Figure supplement 2.** Mp*KNOX1* and Mp*BELL3/4* are dispensable for gametophyte development.

**Figure supplement 3.** Expression of MpKNOX1-GFP by the Mp*KNOX1* promoter complements the karyogamy defects of Mp*knox1* mutants.

**Figure supplement 4.** Sporangium and spore formation in wild-type and Mp*knox1* plants.

**Figure supplement 5.** Mp*KNOX1* is dispensable for sperm differentiation and embryogenesis.

**Figure supplement 6.** Paternally inherited Mp*knox1* does not enhance the zygote arrest phenotype caused by maternally inherited Mp*knox1*.

---

and female pronuclei, approximately a half of zygotes derived from a cross with Mp*bell3/4* males did not exhibit nuclear-enriched GFP signals at 12 HAF (54% and 50 % for Mp*bell3/4-1^{ge}* and Mp*bell3/4-2^{ge}*, respectively; *Figure 6D and G*). Together, these data suggest that paternally inherited Mp*BELL3* and/or Mp*BELL4* contribute to the pronuclear localization of MpKNOX1 after fertilization.

## Discussion

Here, we demonstrated that Mp*KNOX1* is an egg-specific gene in *M. polymorpha*. Mp*KNOX1* is strongly expressed in developing and mature eggs, whereas no expression was detected in gametophytes, sperm, or sporophytes. The egg-specific expression of Mp*KNOX1* is in sharp contrast with the expression pattern of *KNOX1* genes in another model bryophyte, *P. patens*, where all three *KNOX1* genes are strongly expressed in sporophytes to regulate their development (*Sakakibara et al., 2008*). Rather, the egg-specific expression pattern of Mp*KNOX1* is reminiscent of that of *GSM1*, a *KNOX* gene of the unicellular green alga *C. reinhardtii* specifically expressed in *minus* gametes. Upon fertilization, GSM1 forms heterodimers with *plus* gamete-derived BELL protein GSP1 to activate expression of early zygote-specific genes (*Joo et al., 2017*; *Lee et al., 2008*; *Nishimura et al., 2012*). Accordingly, we suspected that MpKNOX1 might function as a gamete-derived zygote activator in *M. polymorpha*. Indeed, a majority of egg cells produced by three independent loss-of-function Mp*knox1* female mutants failed to produce embryos when fertilized with wild-type sperm. By contrast, wild-type eggs fertilized with Mp*knox1* sperm produced normal embryos and spores. The residual embryos produced from Mp*knox1* eggs are unlikely to have formed due to paternal Mp*KNOX1* functions as a comparable number of embryos also arose from homozygous Mp*knox1* zygotes. Rather, it is more likely that the Mp*knox1* alleles generated in our study retained some functionality despite the presence of premature stop codons because a large deletion allele of Mp*knox1* used by another group showed a complete penetrance when transmitted maternally (*Dierschke et al., 2021*). These results indicate that maternal Mp*knox1* has a major, if not exclusive, contribution to embryogenesis.

The parent-of-origin effects of gene alleles can arise at several levels (*Luo et al., 2014*). In some cases, only one parental allele is transcribed in zygotes due to silencing of the other allele. In other cases, gene products such as proteins and small RNA molecules are synthesized in and/or carried over from gametes of a single sex. Our reporter analysis revealed that Mp*KNOX1* is preferentially transcribed in egg cells. By contrast, MpKNOX1-GFP accumulated in both unfertilized and fertilized eggs until 12 HAF and had mostly diminished at 24 HAF. These observations strongly argue for a

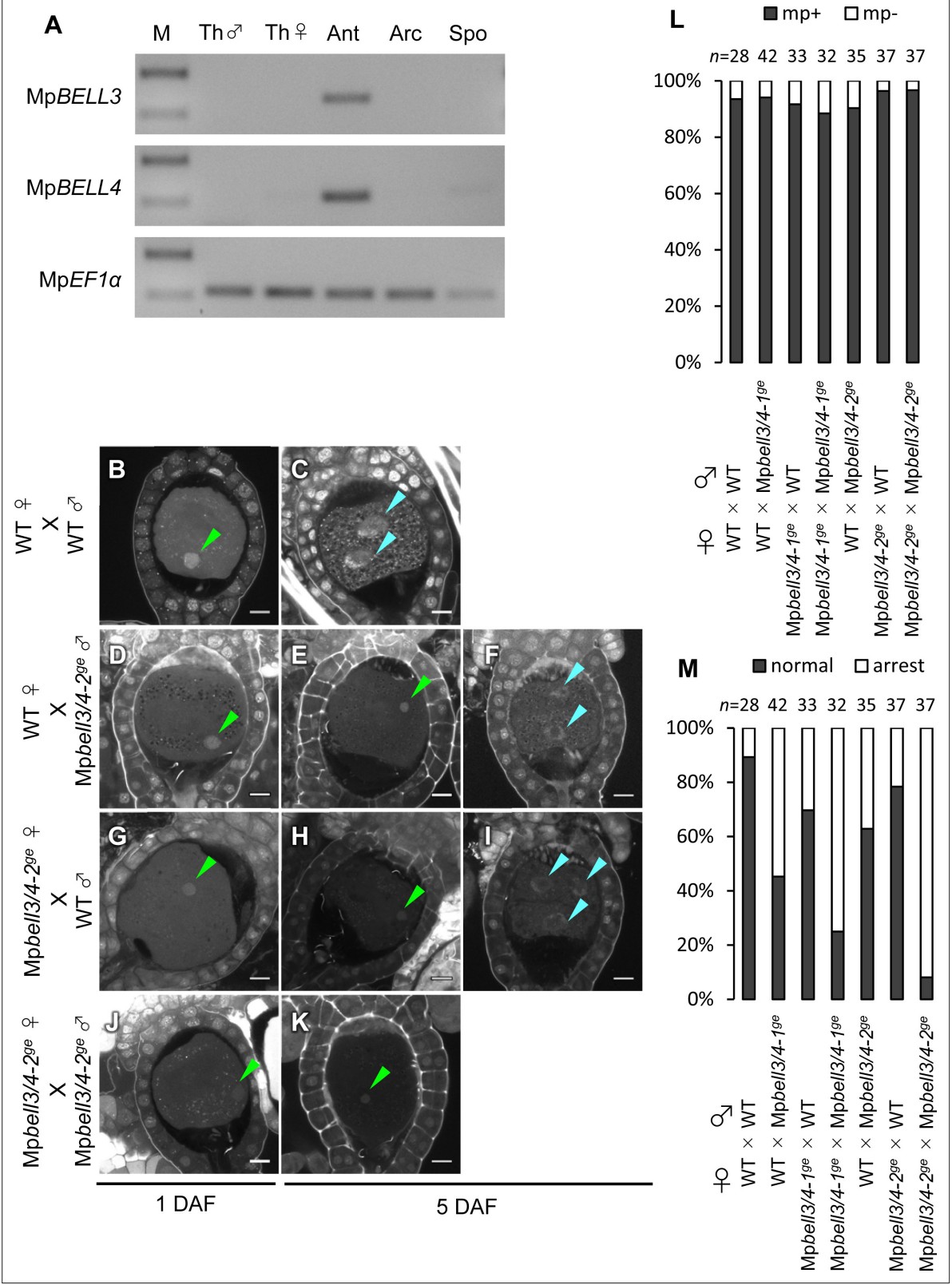

**Figure 5.** Both paternally and maternally inherited Mp*BELL* genes are required for karyogamy. (**A**) RT-PCR analysis indicating that Mp*BELL3* and Mp*BELL4* are specifically expressed in antheridiophores. The lanes are labeled as in *Figure 2A*. Shown is a representative result from the experiments using three independently collected samples each with two technical replicates. See *Figure 4—figure supplement 1A* for the primer position. (**B–K**) Zygotes at 1 day after fertilization (DAF; **B, D, G, J**) and 5 DAF (**C, E, F, H, I, K**) from crosses between a wild-type female and wild-type male (**B, C**) or

*Figure 5 continued on next page*

*Figure 5 continued*

Mp*bell3/4-2^{ge}* male (**D–F**) and a Mp*bell3/4-2^{ge}* female and a wild-type male (**G–I**) or Mp*bell3/4-2^{ge}* male (**J, K**). The presence of male pronuclei (green arrowheads) in zygotes of all genotypes at 1 DAF (**B, D, G, J**) indicates that Mp*BELL3* and Mp*BELL4* are dispensable for plasmogamy. Note that zygotes produced from both or one Mp*bell3/4* parent exhibit a variable degree of karyogamy arrest as visualized by the retention of male pronuclei (green arrowheads) among those starting embryonic division (nuclei labeled with blue arrowheads). Bars, 10 μm. (**L**) A bar graph showing the ratios of zygotes containing male pronuclei (mp+) vs. those not containing male pronuclei (mp-) from indicated crosses at 1 DAF. Numbers of observed zygotes are shown above each bar. (**M**) A bar graph showing the ratios of developed vs. arrested zygotes at 5 DAF from the indicated crosses. Numbers of observed zygotes are shown above each bar.

The online version of this article includes the following figure supplement(s) for figure 5:

**Figure supplement 1.** Mp*BELL3* and Mp*BELL4* are preferentially expressed in antheridiophores.

**Figure supplement 2.** Mp*BELL3* and Mp*BELL4* are dispensable for gamete differentiation.

**Figure supplement 3.** Mp*BELL3* and Mp*BELL4* are dispensable for gamete differentiation.

mechanism in which MpKNOX1 produced in unfertilized eggs acts later in zygotes to initiate embryogenesis. This mode of action of MpKNOX1 is strikingly similar to that proposed for *C. reinhardtii* GSM1 (*Lee et al., 2008*).

How does egg-derived MpKNOX1 promote embryogenesis? We determined that in wild-type *M. polymorpha* karyogamy is completed only after 3–4 DAF. This slow progression of karyogamy is further delayed or arrested when maternal Mp*knox1* is mutated. Observation of nuclear membrane dynamics using an MpSUN-GFP marker showed that in fertilized Mp*knox1* mutant eggs karyogamy was arrested at the nuclear membrane fusion step, not the nuclear migration step. Consistent with the predicted role of MpKNOX1 as a transcription factor, MpKNOX1-GFP localized to both male and female pronuclei by 12 HAF, far before the initiation of nuclear membrane fusion. These observations suggest that gamete-derived MpKNOX1 functions in pronuclei to regulate the expression of gene(s) required for nuclear membrane fusion, thereby activating zygote development.

In both angiosperms and *C. reinhardtii*, the heterodimerization of KNOX and BELL is required for the nuclear translocation of these proteins, which in turn is required for them to regulate gene transcription (*Hay and Tsiantis, 2010*; *Lee et al., 2008*). Consistent with this notion, our genetic and imaging analyses suggested that sperm-derived MpBELL3/4 are required to recruit MpKNOX1 to male and female pronuclei. Our effort of examining MpBELL4 pronuclear localization, however, has been unsuccessful due to a difficulty of obtaining reporter lines, whereas a study by another group indicated interaction between MpKNOX1 and MpBELL4 in tobacco cells (*Dierschke et al., 2021*). Among the five *BELL* genes in the *M. polymorpha* genome, Mp*BELL3* and Mp*BELL4* are specifically expressed in antheridiophores. Mp*bell3/4* mutant males could produce motile sperm, but approximately 30% of wild-type eggs fertilized with Mp*bell3/4* sperm failed to produce embryos compared to ~10% of those fertilized with wild-type sperm. This partial loss of embryogenic ability correlated well with the proportion of zygotes with compromised pronuclear enrichment of MpKNOX1-GFP. These observations, together with other studies on *KNOX/BELL* orthologs, suggest a mechanism in which paternal MpBELL3 and/or MpBELL4 recruit maternal MpKNOX1 to pronuclei.

Interestingly, however, our genetic analyses also indicated that not only paternal but also maternal Mp*BELL3* and/or Mp*BELL4* contribute to karyogamy, even though their expression was not detected in maternal organs or egg cells. A plausible explanation for this observation is that maternally inherited Mp*BELL3* and/or Mp*BELL4* alleles become transcriptionally activated in zygotes and that Mp*BELL3* and/or Mp*BELL4* expression post-fertilization acts to replenish MpBELL proteins to ensure karyogamy with high penetrance. It should be noted, however, that 20–40% of zygotes carrying homozygous Mp*bell3/4* alleles still produce embryos, and this proportion is higher than those carrying maternally inherited Mp*knox1* alone (less than 10%). This lower penetrance of Mp*bell3/4* may be due to redundantly acting Mp*BELL* genes. Transcriptome data indicates weak expression of Mp*BELL1* and Mp*BELL2* in antheridiophores (*Figure 5—figure supplement 1A*) and preferential expression of Mp*BELL5* in archegonia (*Figure 5—figure supplement 1B*). These BELL homologs may play a minor role in promoting karyogamy.

Based on these observations, we propose a two-step model of MpBELL3/4 activity (*Figure 7A*). According to this model, sperm-borne MpBELL3 and MpBELL4 proteins heterodimerize with egg-derived MpKNOX1 upon fertilization and activate the transcription of zygote-specific genes

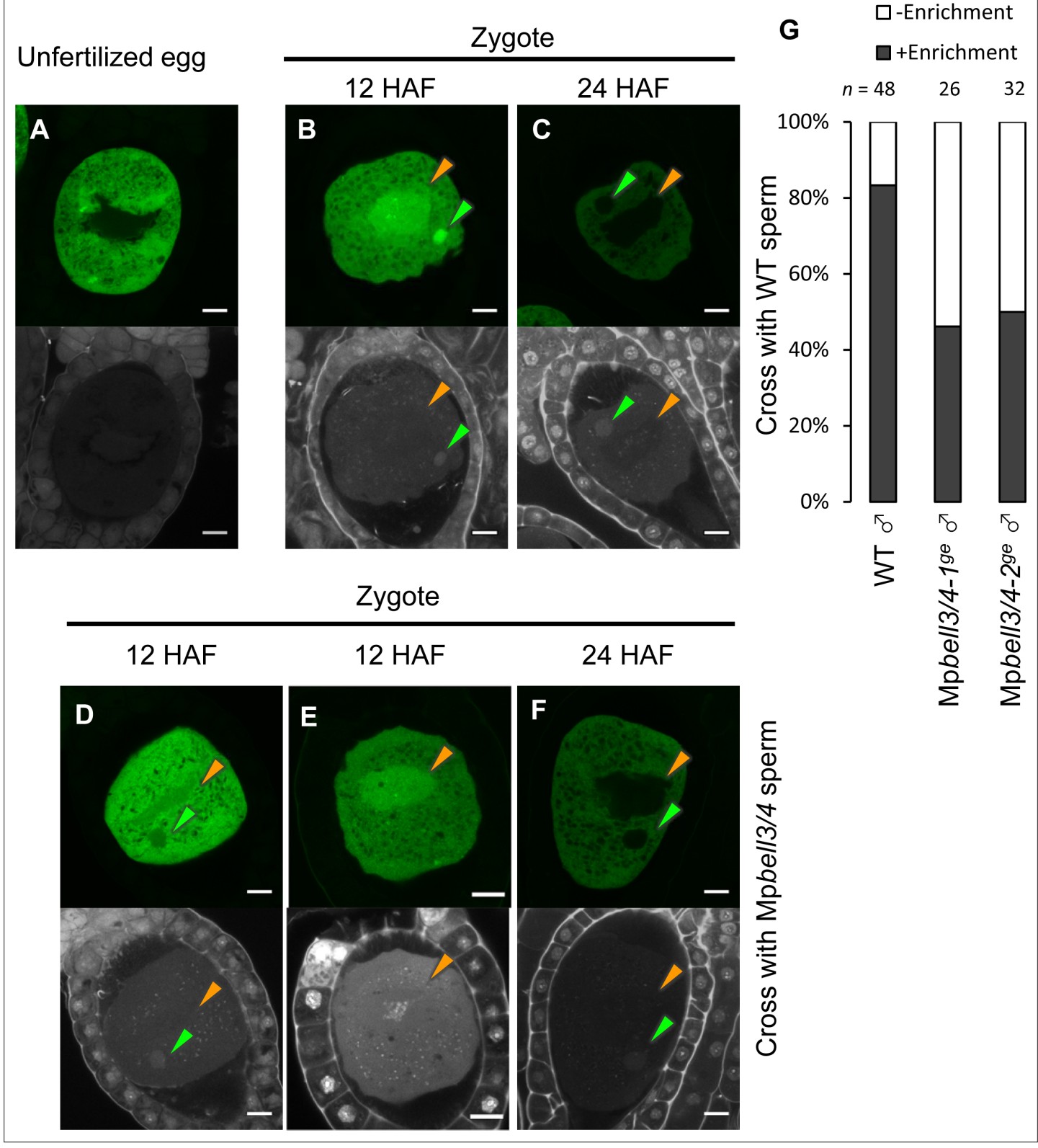

**Figure 6.** MpKNOX1 transiently localizes to female and male pronuclei prior to karyogamy. (**A–F**) GFP (upper panels) and DAPI (lower panels) signals from *gMpKNOX1-GFP/Mpknox1-1^{ge}* eggs (**A**) and zygotes obtained by crossing a *gMpKNOX1-GFP/Mpknox1-1^{ge}* female with a wild-type (**B, C**) or Mp*bell3/4-2^{ge}* (**D–F**) male. Note that before fertilization MpKNOX1-GFP was exclusively localized to the cytosol (**A**). At 12 HAF, MpKNOX1-GFP signals were enriched in female (orange arrowhead) and male (green arrowhead) pronuclei in the wild-type background (**B**). In the absence of paternally

*Figure 6 continued on next page*

*Figure 6 continued*

inherited Mp*BELL3* and Mp*BELL4*, about a half of zygotes did not exhibit nuclear-enriched MpKNOX1-GFP signals at 12 HAF (**D, E**). At 24 HAF, weak and exclusively cytosolic GFP signals were detected in both genotypes (**C, F**). Bars, 10 μm. (**G**) A bar graph showing the ratios of zygotes with and without pronuclei-enriched MpKNOX1-GFP signals from the indicated crosses. Numbers of observed zygotes are shown above each bar.

as well as maternal Mp*BELL3* and/or Mp*BELL4*. MpBELL3 and MpBELL4 produced de novo further activate the expression of zygote-specific genes, including genes required for karyogamy. Such feed-forward regulation would efficiently compensate for the presumably small amounts of MpBELL proteins inherited from the sperm cytosol. Considering that a recent phenotypic analysis of newly isolated *gsm1* and *gsp1* alleles revealed the biparental contribution of GSM1 to karyogamy in *C. reinhardtii* (*Kariyawasam et al., 2019*), our findings further emphasize the similarity of KNOX/BELL-mediated zygote activation between *M. polymorpha* and *C. reinhardtii*.

While our study revealed the striking conservation of KNOX/BELL functions between *M. polymorpha* and *C. reinhardtii*, this finding is somewhat unexpected from a phylogenetic viewpoint because KNOX/BELL proteins in the model bryophyte *P. patens* control sporophyte development, as do KNOX/BELL proteins in angiosperms, as well as egg size (*Sakakibara et al., 2008*; *Horst et al., 2016*; *Ortiz-Ramírez et al., 2017*). Outside the plant kingdom, TALE TFs in yeasts and fungi promote the haploid-to-diploid transition (*Goutte and Johnson, 1988*; *Herskowitz, 1989*; *Kues et al., 1992*, *Spit et al., 1998*). Thus, perhaps the shared zygote-activating functions of KNOX/BELL in *M. polymorpha* and *C. reinhardtii* reflect an ancestral state (*Bowman et al., 2016*). Considering the generally accepted view of bryophyte monophyly (*de Sousa et al., 2019*; *Puttick et al., 2018*), our findings suggest that the functional transition of KNOX/BELLs from zygote activation to sporophyte morphogenesis occurred at least twice during land plant evolution, including once in the bryophyte lineage and once in the tracheophyte lineage (*Figure 7B*).

Our study revealed that the zygote-activating function of KNOX/BELL is conserved between *C. reinhardtii* and *M. polymorpha*, which belong to Chlorophyta and Streptophyta, respectively. These two major green plant lineages were separated more than 700 million years ago (*Becker, 2013*; *Figure 7B*). Interestingly, however, the sex-specific expression patterns of KNOX and BELL in *C. reinhardtii* are opposite to those in *M. polymorpha*. In *C. reinhardtii*, KNOX (GSM1) and BELL (GSP1) are expressed in isogamous *minus* and *plus* gametes (*Lee et al., 2008*), which directly evolved into male and female gametes, respectively, in oogamous (with small motile gametes and large immotile gametes) *Volvox carteri*, primarily by modifying genes acting downstream of the conserved sex-determinant protein MID (*Ferris and Goodenough, 1997*; *Geng et al., 2014*; *Geng et al., 2018*; *Nozaki et al., 2006*; *Figure 7B*). Results from other research groups indicate that the expression specificity of KNOX/BELL is conserved along the *volvocine* lineage (with *minus* and male gametes expressing GSM1, and *plus* and female gametes expressing GSP1; personal communication with Takashi Hamaji [Kyoto University, Japan] and James Umen [Donald Danforth Plant Science Center, MO]). Thus, at least in one Chlorophyta lineage, KNOX and BELL are expressed in male and female gametes, respectively, a situation opposite to that in *M. polymorpha* (*KNOX* for females and *BELL* for males; *Figure 7B*).

In land plants, oogamy likely evolved once as key regulators of gametophytic sexual differentiation, such as FGMYBs for females and DUO POLLEN 1 (DUO1) for males, are shared between *M. polymorpha* and *Arabidopsis* (see *Hisanaga et al., 2019b* for a review; *Figure 7B*). Theoretical analyses predicted that anisogamy evolved through disruptive selection, where an increased volume of one type of gamete favors zygote fitness, allowing the other gamete type to increase in number at the expense of volume (*Parker et al., 1972*). Thus, in ancestral green plants, KNOX/BELL expression specificity did not affect gamete morphology or function. This notion is consistent with our observation that neither Mp*KNOX1* nor Mp*BELL3/4* contribute to gamete development or function in extant *M. polymorpha*.

In summary, our study revealed a critical role of *KNOX* and *BELL* in zygote activation in the model bryophyte *M. polymorpha*. This is in stark contrast with a well-recognized role of *KNOX/BELL* in sporophytic meristem maintenance in angiosperms and another model bryophyte *P. patens*. Rather, striking conservation of KNOX/BELL functions in the promotion of karyogamy across phylogenetically distant *M. polymorpha* and *C. reinhardtii* suggests that functions of

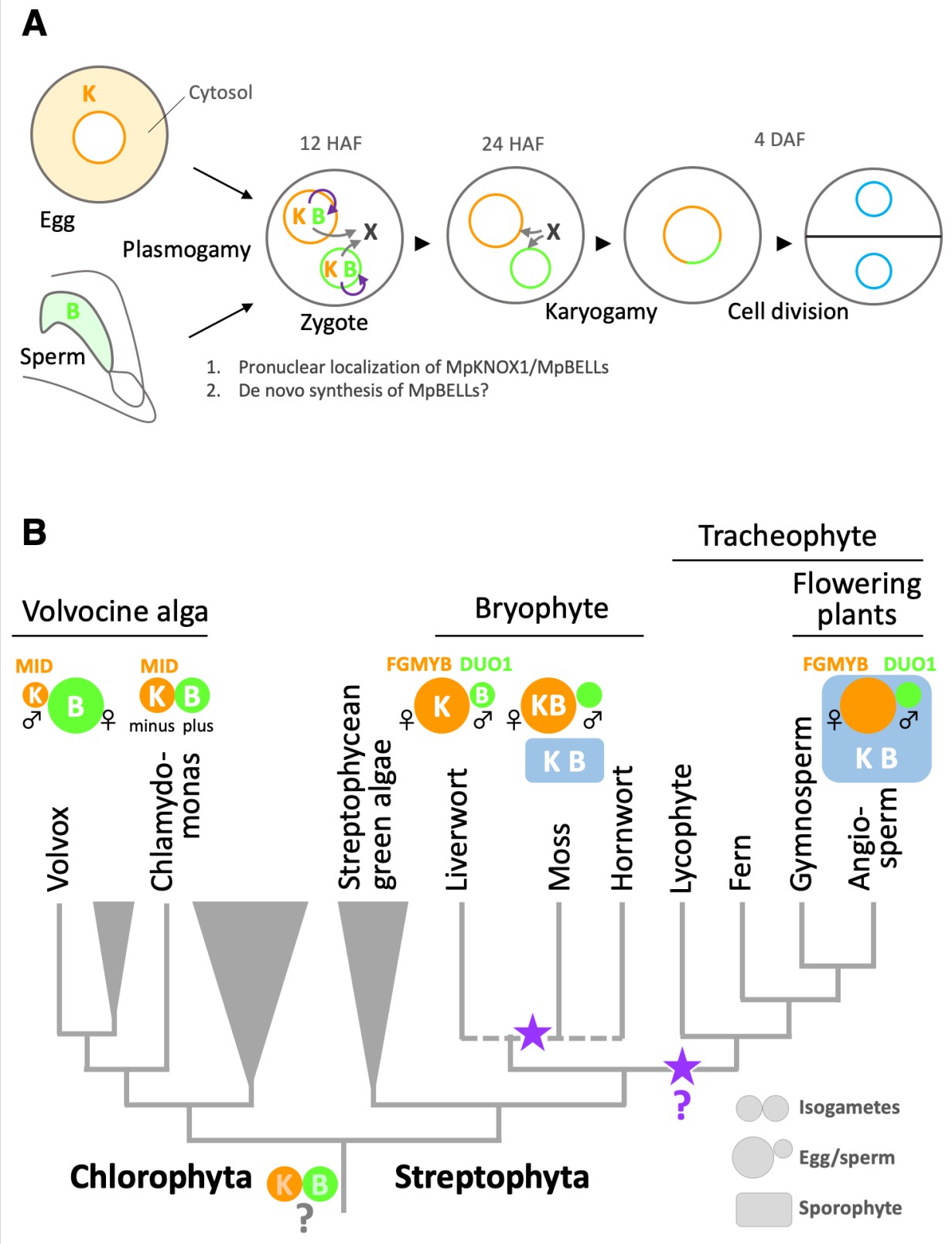

**Figure 7.** Functions and expression patterns of KNOX/BELL transcription factors in green plants. (**A**) Expression patterns of KNOX and BELL proteins and their predicted role in zygote activation in *M. polymorpha*. 'K' and 'B' represent MpKNOX1 and MpBELL protein subunits, respectively. Orange, green, and blue circles represent female pronuclei, male pronuclei, and nuclei of embryo cells, respectively. Purple curved arrows indicate auto-amplification of BELL levels by KNOX/BELL-mediated transcriptional control. X indicates unknown karyogamy-promoting factor(s) whose expression

*Figure 7 continued*

and/or functions are activated by KNOX/BELL-mediated transcription. (**B**) Predicted evolutionary trajectory of KNOX/BELL expression patterns along the green plant lineages. Orange/green circles and blue rectangles represent gametes and sporophyte bodies, respectively. 'K' and 'B' indicate the expression of KNOX and BELL proteins, respectively. Purple stars indicate the predicted positions at which the functional transition of KNOX/BELL from zygote activation to sporophyte morphogenesis occurred. Also indicated are the expression patterns of evolutionarily conserved regulators of sexual differentiation; a male-determinant factor MID of volvocine algae, and female- and male-differentiation factors, FGMYB and DUO1, respectively, of land plants. Note that the KNOX/BELL expression patterns in ancestral plants at the bottom of the tree are an inference.

KNOX/BELL heterodimers shifted from zygote activation to sporophyte development as land plants evolved. This view is consistent with a proposed evolutionary scenario of HD proteins in broader eukaryotic taxa, including fungi and metazoans (*Bowman et al., 2016*).

# Materials and methods

## Key resources table

| Reagent type (species) or resource | Designation | Source or reference | Identifiers | Additional information |
|---|---|---|---|---|
| Gene (*Marchantia polymorpha*) | Mp*KNOX1* | MalpolBase | Mp5g01600 | |
| Gene (*Marchantia polymorpha*) | Mp*BELL1* | MalpolBase | Mp8g18310 | |
| Gene (*Marchantia polymorpha*) | Mp*BELL2* | MalpolBase | Mp4g09650 | |
| Gene (*Marchantia polymorpha*) | Mp*BELL3* | MalpolBase | Mp8g02970 | |
| Gene (*Marchantia polymorpha*) | Mp*BELL4* | MalpolBase | Mp8g07680 | |
| Gene (*Marchantia polymorpha*) | Mp*BELL5* | MalpolBase | Mp5g11060 | |
| Gene (*Marchantia polymorpha*) | Mp*SUN* | MalpolBase | Mp5g02400 | |
| Gene (*Marchantia polymorpha*) | Mp*ECpro* | MalpolBase | Mp5g18000 | |
| Strain, strain background (*Marchantia polymorpha*, male) | Tak-1 | DOI:10.1016/j.cell.2017.09.030 | | Male wild-type strain of *Marchantia* |
| Strain, strain background (*Marchantia polymorpha*, female) | Tak-2 | DOI:10.1016/j.cell.2017.09.030 | | Female wild-type strain of *Marchantia* |
| Recombinant DNA reagent | pMpGE_En03 | Addgene | RRID:Addgene_71535 | Entry plasmid for gRNA |
| Recombinant DNA reagent | pMpGE010 | Addgene | RRID:Addgene_71536 | Destination plasmid for CRISPR/Cas9 |
| Recombinant DNA reagent | pMpGE_En04 | DOI:10.15252/embj.2018100240 | | Entry plasmid for gRNA |
| Recombinant DNA reagent | pBC-GE12 | DOI:10.15252/embj.2018100240 | | Entry plasmid for gRNA |
| Recombinant DNA reagent | pBC-GE23 | DOI:10.15252/embj.2018100240 | | Entry plasmid for gRNA |
| Recombinant DNA reagent | pBC-GE34 | DOI:10.15252/embj.2018100240 | | Entry plasmid for gRNA |
| Recombinant DNA reagent | pMpGE017 | DOI:10.15252/embj.2018100240 | | Destination plasmid for Cas9 nickase |
| Recombinant DNA reagent | pMpGE010_MpKNOX1ge | This paper | | Plasmid to create Mp*knox1* mutants |
| Recombinant DNA reagent | pMpGE017_MpBELL4ge-MpBELL3ge | This paper | | Plasmid to create Mp*bell3/4* mutants |
| Recombinant DNA reagent | pMpSL30 | DOI:10.15252/embj.2018100240 | | Destination plasmid |
| Recombinant DNA reagent | pMpSL30_MpKNOX1pro-H2B-GFP-3'MpKNOX1 | This paper | | Mp*KNOX1* promoter reporter |

*Continued on next page*

*Continued*

| Reagent type (species) or resource | Designation | Source or reference | Identifiers | Additional information |
|---|---|---|---|---|
| Recombinant DNA reagent | pMpSL30_gMpKNOX1-GFP | This paper | | Mp*KNOX1* complementation |
| Recombinant DNA reagent | pMpSL30_ ECpro-MpSUN-GFP | This paper | | Egg cell-specific nuclear envelop marker |
| Genetic reagent (*Marchantia polymorpha*) | Mp*knox1-1^ge* | This paper | | Mp*knox1* CRISPR mutant obtained from sporeling transformation |
| Genetic reagent (*Marchantia polymorpha*) | Mp*knox1-2^ge* | This paper | | Mp*knox1* CRISPR mutant obtained from sporeling transformation |
| Genetic reagent (*Marchantia polymorpha*) | Mp*knox1-3^ge* | This paper | | Mp*knox1* CRISPR mutant obtained from sporeling transformation |
| Genetic reagent (*Marchantia polymorpha*) | Mp*bell3/4-1^ge* | This paper | | Mp*bell3/4* CRISPR mutant obtained from sporeling transformation |
| Genetic reagent (*Marchantia polymorpha*) | Mp*bell3/4-2^ge* | This paper | | Mp*bell3/4* CRISPR mutant obtained from sporeling transformation |
| Genetic reagent (*Marchantia polymorpha*) | *MpKNOX1pro:H2B-GFP/WT* ♀ | This paper | | pMpSL30_MpKNOX1pro-H2B-GFP-3'MpKNOX1 was transformed into Tak-2 |
| Genetic reagent (*Marchantia polymorpha*) | *gMpKNOX1-GFP/Mpknox1-1^ge*♀ | This paper | | pMpSL30_gMpKNOX1-GFP was transformed into Mp*knox1-1^ge* |
| Genetic reagent (*Marchantia polymorpha*) | *ECpro:MpSUN-GFP* | This paper | | pMpSL30_ ECpro-MpSUN-GFP was transformed into Tak-2 or Mp*knox1-2^ge* |
| Genetic reagent (*Marchantia polymorpha*) | Mp*rkd-1* | DOI:10.1016/j.cub.2016.05.013 | | |
| Genetic reagent (*Marchantia polymorpha*) | Mp*rkd-3* | DOI:10.1016/j.cub.2016.05.013 | | |
| Commercial assay or kit | RNeasy Plant Mini Kit | Qiagen | 74904 | |
| Commercial assay or kit | TruSeq RNA Sample Prep Kit v2 | Illumina | RS-122-2001 | |
| Commercial assay or kit | Gateway LR Clonase II Enzyme mix | Thermo Fisher Scientific | 11791020 | |
| Software, algorithm | TopHat ver. 2.0.14 | doi:10.1093/bioinformatics/btp120 | RRID:SCR_013035 | |
| Software, algorithm | DESeq2 | doi:10.1186/s13059-014-0550-8 | RRID:SCR_015687 | |
| Software, algorithm | TCC | DOI:10.1186/1471-2105-14-219 | RRID:SCR_001779 | |

## Plant materials

Marchantia polymorpha L. subsp. *ruderalis* accessions Takaragaike 1 (Tak-1) and Takaragaike 2 (Tak-2; *Ishizaki et al., 2016*) were used as the wild-type male and female, respectively. Plants were cultured on half-strength Gamborg's B5 medium solidified with 1% (w/v) agar under continuous white light at 22 °C. To induce reproductive development, 10-day-old thalli were transferred to a pot containing vermiculite and grown under white light supplemented with far-red illumination generated by LED (VBL-TFL600-IR730; IPROS Co., Tokyo, Japan).

## Archegonium sampling, RNA extraction, and Illumina sequencing

Mature archegonia were manually dissected from wild-type archegoniophores and divided into four pools, each composed of approximately 1000 archegonia. Mature archegonia of Mp*rkd-1* and Mp*rkd-3* (*Koi et al., 2016*) were collected and each divided into two pools of approximately 1000 archegonia. Total RNA was extracted from the samples with an RNeasy Plant Mini Kit (Qiagen, Venlo, Netherlands) according to the manufacturer's protocol. The quality and quantity of the RNA were evaluated using a Qubit 2.0 Fluorometer (Life Technologies, Carlsbad, CA) and a RNA6000 Nano Kit (Agilent Technologies, Santa Clara, CA). Sequence libraries were constructed with a TruSeq RNA Sample Prep Kit v2 (Illumina, San Diego, CA) according to the manufacturer's protocol. The quality of each library was examined using a Bioanalyzer with High Sensitivity DNA Kit (Agilent Technologies) and a KAPA Library

Quantification Kit for Illumina (Roche Diagnostics, Basel, Switzerland). An equal amount of each library was mixed to generate a 2 nM pooled library. Next-generation sequencing was performed using the HiSeq 1500 platform (Illumina) to generate 126-nt single-end data. Sequence data have been deposited at the DDBJ BioProject and BioSample databases under accession numbers PRJDB9329 and SAMD00205647-SAMD00205654, respectively.

## Data analysis

Read data were mapped to the genome sequence of *M. polymorpha* v3.1 using TopHat ver. 2.0.14 (*Trapnell et al., 2009*) with default parameters. FPKM values were calculated using the DESeq2 package in R (*Love et al., 2014*). Differentially expressed genes (DEGs) were identified using the TCC package in R (*Sun et al., 2013*) with a criterion of false discovery rate (FDR) < 0.01. Candidate egg-specific genes were identified by filtering the DEGs with a threshold of Mp*rkd*/WT ratio <−3 and FPKM in WT > 1.

## Semi in vitro culture and genetic crossing

Mature archegoniophores and antheridiophores were separated from thalli and collected into a 5 mL plastic tube containing 3 mL of water. Following co-culture for 1 hr under white light at 22 °C, the archegoniophores were transferred to new 5 mL plastic tubes containing 3 mL of water and cultured under white light at 22 °C prior to observation.

## Microscopy

To observe MpSUN-GFP expression, archegonia were excised under a dissecting microscope and mounted in half-strength Gamborg's B5 liquid medium. To observe pronuclei, archegoniophores were dissected and soaked in PFA fixative solution (4% [w/v] paraformaldehyde, 1 µg/mL DAPI, and 0.01% [v/v] Triton X-100 in 1× PBS buffer). The samples were briefly vacuum-infiltrated four times and incubated for 1 hr at room temperature with gentle shaking. The samples were washed twice with 1× PBS and cleared by incubating in ClearSee solution containing 1 µg/mL DAPI for 2–3 days (*Kurihara et al., 2015*). The cleared samples were observed under a Nikon C2 confocal laser-scanning microscope (Nikon Instech, Tokyo, Japan). Sperm cells were stained with DAPI as described previously (*Hisanaga et al., 2019a*).

## RT-PCR

RNA extraction, cDNA synthesis, and RT-PCR were performed as described previously (*Hisanaga et al., 2019a*) using the primer sets listed in *Supplementary file 2*.

## DNA constructs

The plasmids used in this study were constructed using the Gateway cloning system (*Ishizaki et al., 2015*), the SLiCE method (*Motohashi, 2015*), or Gibson assembly (*Gibson et al., 2009*). The primers used for DNA construction are listed in *Supplementary file 2*.

### pMpGE010_MpKNOX1ge

A DNA fragment producing Mp*KNOX1*-tageting gRNAs was prepared by annealing a pair of synthetic oligonucleotides (MpKNOX1ge01Fw/MpKNOX1ge01Rv). The fragment was inserted into the BsaI site of pMpGE_En03 (cat. no. 71535, Addgene, Cambridge, MA) to yield pMpGE_En03-MpKNOX1ge, which was transferred into pMpGE010 (cat. no. 71536, Addgene) (*Sugano et al., 2018*) using the Gateway LR reaction (Thermo Fisher Scientific, Waltham, MA) to generate pMpGE010_MpKNOX1ge.

### pMpGE017_MpBELL4ge-MpBELL3ge

To construct a plasmid to disrupt Mp*BELL3* and Mp*BELL4* simultaneously, four oligonucleotide pairs (MpBELL4-ge1-Fw/MpBELL4-ge1-Rv, MpBELL4-ge2-Fw/MpBELL4-ge2-Rv, MpBELL3-ge3-Fw/MpBELL3-ge3-Rv, and MpBELL3-ge4-Fw/MpBELL3-ge4-Rv) were annealed and cloned into pMpGE_En04, pBC-GE12, pBC-GE23, and pBC-GE34 to yield pMpGE_En04-MpBELL4-ge1, pBC-GE12-MpBELL4-ge-2, pBC-GE23-MpBELL3-ge3, and pBC-GE34-MpBELL3-ge4, respectively. These four plasmids were assembled via BglI restriction sites and ligated to yield pMpGE_En04-MpBELL4-ge12-MpBELL3-ge34. The resulting DNA fragment containing four

MpU6promoter-gRNA cassettes was transferred into pMpGE017 using the Gateway LR reaction to yield pMpGE017_MpBELL4-ge12-MpBELL3-ge34.

### pMpSL30_MpKNOX1pro-H2B-GFP-3′MpKNOX1

A 6.3 kb genomic fragment spanning the 5.3 kb 5′ upstream sequence plus the 1 kb 5′-UTR of MpKNOX1 was amplified from Tak-2 genomic DNA using the primers H-MpKNOX1pro-Fw and SmaI-MpKNOX1pro-Rv. A vector backbone containing the GFP-coding sequence was prepared by digesting pAN19_SphI-35S-lox-IN-lox-NosT-SmaI-GFP-NaeI (a kind gift from Dr. Shunsuke Miyashima) with SphI and SmaI. The two fragments were assembled using the SLiCE reaction to yield pAN19_MpKNOX1pro-SmaI-GFP-NaeI. The histone H2B-coding sequence from *Arabidopsis* was amplified from the pBIN41_DUO1pro-H2B-YFP-nos vector (Hisanaga, unpublished) using the primers KNOXp-H2B-Fw and GFP-H2B-Rv. The fragment was inserted into the SmaI site of pAN19_MpKNOX1pro-SmaI-GFP-NaeI by the SLiCE reaction, yielding pAN19_MpKNOX1pro-H2B-GFP-NaeI. A 4 kb fragment containing the 0.5 kb 3′-UTR plus 3.5 kb 3′-flanking sequences of MpKNOX1 was amplified from Tak-2 genomic DNA using the primers G-MpKNOX1ter-Fw and E-MpKNOX1ter-Rv. The fragment was inserted into the NaeI site of pAN19_MpKNOX1pro-H2B-GFP-NaeI by the SLiCE reaction, yielding pAN19_MpKNOX1pro-H2B-GFP-3′MpKNOX1. The MpKNOX1pro-H2B-GFP-3′MpKNOX1 fragment was excised from pAN19_MpKNOX1pro-H2B-GFP-3′MpKNOX1 by digestion with AscI and inserted into pMpSL30 (*Hisanaga et al., 2019a*) to yield pMpSL30_MpKNOX1pro-H2B-GFP-3′MpKNOX1.

### gMpKNOX1-GFP

A 3.4 kb genomic fragment spanning the entire exon and intron region of MpKNOX1 was amplified from Tak-2 genomic DNA using the primers gMpKNOX1-Fw and gMpKNOX1-Rv. The fragment was inserted into the SmaI site of pAN19_MpKNOX1pro-SmaI-GFP-NaeI by the SLiCE reaction to yield pAN19_MpKNOX1pro-MpKNOX1-GFP-NaeI. A 4 kb fragment containing the 0.5 kb 3′-UTR and 3.5 kb 3′-flanking sequences of MpKNOX1 was amplified from Tak-2 genomic DNA using the primers G-MpKNOX1ter-Fw and E-MpKNOX1ter-Rv. The fragment was inserted into the NaeI site of pAN19_MpKNOX1pro-MpKNOX1-GFP-NaeI by the SLiCE reaction, yielding pAN19_gMpKNOX1-GFP. The gMpKNOX1-GFP fragment was excised from pAN19_gMpKNOX1-GFP by digestion with AscI and inserted into pMpSL30 to yield pMpSL30_gMpKNOX1-GFP.

### ECpro:MpSUN-GFP

A 5 kb genomic fragment spanning the 3.4 kb 5′ upstream and 1.6 kb 5′-UTR sequences of Mp5g18000 was amplified from Tak-2 genomic DNA using the primers H-ECpro-Fw and G-SpeI-ECpro-Rv. The fragment was inserted into the SpeI site of pAN19_SpeI-GFP-NaeI by the SLiCE reaction, yielding pAN19_ECpro-SpeI-GFP-NaeI. The 1.5 kb MpSUN (Mp5g02400)-coding sequence was amplified from a cDNA library of *M. polymorpha* using the primers E-MpSUN-Fw and G-MpSUN-Rv. The fragment was inserted into the SpeI site of pAN19_ECpro-SpeI-GFP-NaeI by the SLiCE reaction, yielding pAN19_ECpro-MpSUN-GFP. The ECpro-MpSUN-GFP fragment was excised from pAN19_ECpro-MpSUN-GFP by digestion with AscI and inserted into pMpSL30 to yield pMpSL30_ECpro-MpSUN-GFP.

## Generation of transgenic *M. polymorpha*

Genome editing constructs were introduced into *M. polymorpha* sporelings as described previously (*Ishizaki et al., 2008*). Other constructs were introduced into regenerating thalli (*Kubota et al., 2013*) or gemmae using the G-AgarTrap method (*Tsuboyama et al., 2018*).

## Acknowledgements

We thank Masako Kanda for technical assistance and Shunsuke Miyashima for DNA materials. This work was supported by MEXT KAKENHI grants 17J08430 to TH, 25113007 to KN, 17H05841 and 18K06285 to SY, and 25113009 and 17H07424 to TK. TH was supported by a JSPS Fellowship for Young Scientists and a funding from the European Union's Framework Programme for Research and Innovation Horizon 2020 (2014–2020) under the Marie Curie Skłodowska Grant Agreement Nr. 847548.

## Additional information

### Funding

| Funder | Grant reference number | Author |
|---|---|---|
| Japan Society for the Promotion of Science | KAKENHI 17J08430 | Tetsuya Hisanaga |
| Ministry of Education, Culture, Sports, Science and Technology | KAKENHI 25113007 | Keiji Nakajima |
| Japan Society for the Promotion of Science | KAKENHI 18K06285 | Shohei Yamaoka |
| Ministry of Education, Culture, Sports, Science and Technology | KAKENHI 17H05841 | Shohei Yamaoka |
| Ministry of Education, Culture, Sports, Science and Technology | KAKENHI 25113009 | Takayuki Kohchi |
| Japan Society for the Promotion of Science | KAKENHI 17H07424 | Takayuki Kohchi |
| European Commission | Marie Sklodowska-Curie grant agreement 847548 | Tetsuya Hisanaga |

The funders had no role in study design, data collection and interpretation, or the decision to submit the work for publication.

### Author contributions

Tetsuya Hisanaga, Conceptualization, Formal analysis, Funding acquisition, Investigation, Methodology, Visualization, Writing - original draft; Shota Fujimoto, Yihui Cui, Katsutoshi Sato, Investigation; Ryosuke Sano, Data curation, Formal analysis, Software; Shohei Yamaoka, Funding acquisition, Investigation, Writing - review and editing; Takayuki Kohchi, Funding acquisition, Project administration; Frédéric Berger, Supervision, Writing - review and editing; Keiji Nakajima, Conceptualization, Funding acquisition, Project administration, Supervision, Visualization, Writing - original draft

### Author ORCIDs

Tetsuya Hisanaga http://orcid.org/0000-0002-2834-7044
Shohei Yamaoka http://orcid.org/0000-0003-4154-9967
Takayuki Kohchi http://orcid.org/0000-0002-9712-4872
Frédéric Berger http://orcid.org/0000-0002-3609-8260
Keiji Nakajima http://orcid.org/0000-0002-1580-3354

### Decision letter and Author response

Decision letter https://doi.org/10.7554/eLife.57090.sa1
Author response https://doi.org/10.7554/eLife.57090.sa2

## Additional files

### Supplementary files

• Supplementary file 1. List of genes with changed mRNA levels in Mp*rkd* archegonia. Genes with reduced and increased expression levels in Mp*rkd* archegonia are listed in sheet 1 and sheet 2, respectively.

• Supplementary file 2. Primers used in this study. Lowercase and uppercase letters indicate synthetic adaptor and target DNA sequences, respectively.

• Transparent reporting form

### Data availability

Sequence data have been deposited at the DDBJ BioProject and BioSample databases under accession numbers PRJDB9329 and SAMD00205647-SAMD00205654, respectively.

The following dataset was generated:

| Author(s) | Year | Dataset title | Dataset URL | Database and Identifier |
|---|---|---|---|---|
| Hisanaga T, Sato K, Sano R, Yamaoka S, Kohchi T, Nakajima K | 2020 | Transcriptome analysis of archegonia in Marchantia wild type and Mprkd mutant | https://www.ebi.ac.uk/ena/browser/view/PRJDB9329 | EBI European Nucleotide Archive, PRJDB9329 |

The following previously published datasets were used:

| Author(s) | Year | Dataset title | Dataset URL | Database and Identifier |
|---|---|---|---|---|
| DOE Joint Genome Institute | 2014 | Marchantia polymorpha strain:Takaragaike-1 (male) Takaragaike-1 (female, Takaragaike-2 was back-crossed four times to Takaragaike-1) (liverwort) | https://www.ncbi.nlm.nih.gov/bioproject/?term=PRJNA251267 | NCBI BioProject, PRJNA251267 |

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
