## [Decision Letter]

**Acceptance summary:**

Hisanaga et al., analyze the roles of the homeodomain transcription factors KNOX1 and BELL in the liverwort Marchantia polymorpha, a model species suitable to study evolution of sexual reproduction in plants. They ask whether TALE class KNOX/BELL heterodimeric transcription factors play a critical role in zygote development of Marchantia polymorpha. By analyzing transcriptomes of the egg-defective MpRKD mutant, the authors identified the single KNOX1 gene that is female gamete-specific and two BELL family genes that are male gamete-specific. The maternal (but not paternal) MpKNOX1 is required for activating zygote development, specifically by promoting male and female pronuclear fusion and successful fertilization. This function of MpKNOX1 appears to depend on maternal and maternal MpBELL genes. They established an in vitro procedure to examine the details of the Marchantia zygote development in a synchronized manner. They used a CRISPR-based knockout approach to show that MpKNOX1 and MpBELL3/4 are required for distinct cell biological events in the fertilized eggs: MpKNOX1 for fusion of pro-nuclei and MpBELL3/4 for the nuclear localization of egg-derived MpKNOX1. In contrast, KNOX and BELL proteins in land plants function during the sporophyte generation to control later stages of development, such as meristem maintenance and organogenesis. This led the authors to propose 'deep ancestry' of gamete-directed zygote activation by KNOX/BELL, conserved between green algae and land plants. The observed configuration of KNOX/BELL in Marchantia is similar to that in *Chlamydomonas reinhardtii*, a green alga, and apparently distinct from that in *Physcomitrium patens*. Therefore, the manuscript provides novel and important evidence that zygote activation is the ancestral role of KNOX/BELL transcription factors, and suggests that the meristem maintenance role of these proteins evolved secondarily in land plants.

**Decision letter after peer review:**

Thank you for submitting your article "Deep evolutionary origin of gamete-directed zygote activation by KNOX/BELL transcription factors in green plants" for consideration by *eLife*. Your article has been reviewed by 3 peer reviewers, including Sheila McCormick as Reviewing Editor and Reviewer #1, and the evaluation has been overseen by Christian Hardtke as the Senior Editor.

The reviewers have discussed the reviews with one another and the Reviewing Editor has drafted this decision to help you prepare a revised submission.

As the editors have judged that your manuscript is of interest, but as described below that additional experiments are required before it is published, we would like to draw your attention to changes in our revision policy that we have made in response to COVID-19 (https://elifesciences.org/articles/57162). First, because many researchers have temporarily lost access to the labs, we will give authors as much time as they need to submit revised manuscripts. We are also offering, if you choose, to post the manuscript to bioRxiv (if it is not already there) along with this decision letter and a formal designation that the manuscript is "in revision at eLife". Please let us know if you would like to pursue this option. (If your work is more suitable for medRxiv, you will need to post the preprint yourself, as the mechanisms for us to do so are still in development.)

Summary:

Hisanaga et al., describe the analysis of the roles of the homeodomain transcription factors KNOX1 and BELL in the liverwort Marchantia polymorpha, a model species suitable to study evolution of sexual reproduction in plants. They ask whether TALE class KNOX/BELL heterodimeric transcription factors play a critical role in zygote development of Marchantia polymorpha. By analyzing transcriptomes of the egg-defective MpRKD mutant, the authors identified the single KNOX1 gene that is female gamete-specific and two BELL family genes that are male gamete-specific. The maternal (but not paternal) MpKNOX1 is required for activating zygote development, specifically by promoting male and female pronuclear fusion and successful fertilization. This function of MpKNOX1 appears to depend on maternal and maternal MpBELL genes.They established an in vitro procedure to examine the details of the Marchantia zygote development in a synchronized manner. They used a CRISPR-based knockout approach to show that MpKNOX1 and MpBELL3/4 are required for distinct cell biological events in the fertilized eggs: MpKNOX1 for fusion of pro-nuclei and MpBELL3/4 for the nuclear localization of egg-derived MpKNOX1. In contrast, KNOX and BELL proteins in land plants function during the sporophyte generation to control later stages of development, such as meristem maintenance and organogenesis. This led the authors to propose 'deep ancestry' of gamete-directed zygote activation by KNOX/BELL, conserved between green algae and land plants. The observed configuration of KNOX/BELL in Marchantia is similar to *Chlamydomonas reinhardtii*, a green alga, and apparently distinct from that in *Physcomitrium patens*. Therefore, the manuscript provides novel and important evidence that zygote activation is the ancestral role of KNOX/BELL transcription factors, and suggests that the meristem maintenance role of these proteins evolved secondarily in land plants.

Essential revisions:

GFP lines to localize MpBELL, additional crosses. See details below.

1) The authors used time-resolved microscopic observations of subcellular dynamics in zygotes (nuclear fusion and cell wall deposition) using an elegant in vitro fertilization protocol that allows synchronization of fertilization and detailed observation of the subcellular events occurring during fertilization. The cell biology approach was used together with a range of loss of function mutants of maternal/paternal MpKNOX1 and MpBELL3,4, and MpKNOX1-GFP lines to follow subcellular protein localization. The paper presents convincing evidence for the requirement of maternal KNOX1 and paternal/maternal MpBELL3,4 for pronuclear fusion and successful fertilization. The evidence for the need of paternal MpBELL3,4 to the pronuclear localization of MpKNOX1 protein after fertilization is also compelling. However, it is somewhat disappointing not to have more information about the interactions between KNOX and BELL. One obvious inference from the results of the cross wt female x MpBELL3,4 male is that nuclear localization and/or nuclear retention requires heterodimerization (as is the case in *Chlamydomonas*). In the absence of in vitro or in vivo interaction assays, they could have exploited their mutant and GFP lines to further characterize the links between KNOX1 and BELL3,4. For example, they could have used GFP lines to localize MpBELL to the pronuclei, as they did for MpKNOX. Is there a reason why this is not possible? They could also have done the opposite experiment, i.e., use Mpknox1 mutant females x MpBELL-GFP lines to show (presumably) that BELL is absent from the pronuclei. This is what is expected if their model (Figure 7) is right, and it would be an important result to add to the manuscript.

2) Both maternal and paternal BELL3,4 are needed for fusion of pronuclei, but a considerable number of zygotes still went through successful fertilization in the absence of maternal/paternal BELL (78% and 63% respectively) (Line 268-276). This suggests that these "successful zygotes" use their 'uniparental' BELL to achieve karyogamy (or that other BELL proteins take over?). Therefore, it is expected that in the cross of MpKNOX-GFP female x Mpbell3,4 male (which retain GFP-KNOX in the cytosol) a large proportion of these zygotes would also achieve fertilization and therefore exhibit nuclear-localised KNOX (and BELL). However, it appears that zygotes from this cross do not have nuclearly-localized KNOX (and the authors do not mention how many replicate zygotes were used, an important piece of information that needs to be added). This is quite surprising. One would expect that more than half of the GFP lines would have a GFP signal from KNOX (and from BELL as well, if GFP-BELL was looked at) in the pronuclei. If none of the zygotes arising from this cross have GFP signal on the pronuclei, it suggests that successful fertilization can be achieved even in the absence of nuclear localization of KNOX? Please clarify.

3. RNA-seq results of the Mprkd mutant (Figure 1) were not disclosed other than expression of MpKNOX1. Suggest including a table for the genes that were significantly more expressed in the wild type than in the Mprkd mutant archegonia.

4. MpKNOX1 is a single gene in the highly conserved KNOX1 family, predicting severe defects in the Mpknox1 mutant. As acknowledged by the authors, two independent Mpknox1-1ge and Mpknox1-2ge mutants showed incomplete defects and produced mature spores, indicating successful completion of sporophyte development. Results in Figure 4 need to be improved by providing quantitative details of zygote development in more than one knockout line, including fusion of pro-nuclei, # of capsules/sporangia, the germination rate of tetraspores,.

5. The pronuclear fusion defect of Mpknox1-ko is one of the most critical observations, leading to the authors' conclusion that KNOX1 is required for zygote activation. Nuclear fusion itself is, however, not necessarily a pre-requisite for nor dependent on zygote activation. It is, therefore, important to report the pronuclear fusion rate at time points after 5 DAF. Provide quantitative analysis of pronuclear fusion and report whether zygote development proceeds with or without nuclear fusion.

6. On the other hand, the leaky phenotypes of Mpknox1 mutants might be due to MpKNOX1 expression from the male genome in the zygotes of wild type male X Mpknox1-ko female, analogous to the Mpbell3/4-ko case presented in Figure 5. Since Figure 4 – Sup3 reported the phenotype of Mpknox1-1ge males, it is possible to examine Mpknox1-ko male X Mpknox1-ko female? This is critical for testing the two scenarios presented by the authors: leaky expression of MpKNOX1 and genetic redundancy.

7. Mpbell3/4 double mutants display leaky phenotypes. Incomplete arrest in early-stage zygote development might suggest genetic redundancy. Since the Marchantia genome possesses five BELL genes, comments about potential redundancy due to the other BELL family genes should be included. For instance, changes in MpKNOX1 localization is presented in support for a critical role of MpBELL3/4. Figure 6D shows residual MpKNOX1GFP signals in two nuclei at 12 HAF, and exclusion or disappearance of the signals from the nuclei at 24 HAF is evident in Figures 6C and 6E, suggesting that KNOX1 movement might occur in Mpbell3/4. This movement could be explained by MpBELL3/4 produced from the female nucleus or by redundant BELL.

8. Zygote cell division at 5 DAF and nuclear translocation of MpKNOX1 in 12 HAF zygotes were reported as the primary defects of the Mpbell3/4 double mutants. Provide a quantitative assessment of MpKNOX1 nuclear localization between the BELL mutants and wild type. Also, provide details on their other phenotypes, such as whether the resulting zygotes develop into a partial embryo or produce spores. Such quantitative assessment is necessary for a complete evaluation of the MpBELL3/4 functions.

9. While two Mpbell3/4 mutant lines were reported (Figure 4 – sup 2), Figures 5 and 6 include the results from only single Mpbell3/4 in the absence of genetic complementation. Provide the phenotypes of both lines for proper assessment of the Mpbell3/4 double knockouts.

---

## [Author Response]

Essential revisions:GFP lines to localize MpBELL, additional crosses. See details below.1) The authors used time-resolved microscopic observations of subcellular dynamics in zygotes (nuclear fusion and cell wall deposition) using an elegant in vitro fertilization protocol that allows synchronization of fertilization and detailed observation of the subcellular events occurring during fertilization. The cell biology approach was used together with a range of loss of function mutants of maternal/paternal MpKNOX1 and MpBELL3,4, and MpKNOX1-GFP lines to follow subcellular protein localization. The paper presents convincing evidence for the requirement of maternal KNOX1 and paternal/maternal MpBELL3,4 for pronuclear fusion and successful fertilization. The evidence for the need of paternal MpBELL3,4 to the pronuclear localization of MpKNOX1 protein after fertilization is also compelling. However, it is somewhat disappointing not to have more information about the interactions between KNOX and BELL. One obvious inference from the results of the cross wt female x MpBELL3,4 male is that nuclear localization and/or nuclear retention requires heterodimerization (as is the case in Chlamydomonas). In the absence of in vitro or in vivo interaction assays, they could have exploited their mutant and GFP lines to further characterize the links between KNOX1 and BELL3,4. For example, they could have used GFP lines to localize MpBELL to the pronuclei, as they did for MpKNOX. Is there a reason why this is not possible? They could also have done the opposite experiment, i.e., use Mpknox1 mutant females x MpBELL-GFP lines to show (presumably) that BELL is absent from the pronuclei. This is what is expected if their model (Figure 7) is right, and it would be an important result to add to the manuscript.

In order to address this important issue, we have been working to generate MpBELL4-FP lines, but we encountered problems. To express MpBELL4-FP, we used a 5.8-kb promoter of Mp*BELL4*, as it is shown to drive antheridia-specific expression in the accompanying paper by Dierschke et al., We fused this promoter fragment with the MpBELL4-coding sequence, an mScarlet-coding sequence and 0.7-kb MpBELL4 3’ franking sequence. We introduced the construct into Mp*bell3/4-1^ge^* males and obtained seven independent lines. We observed antheridia and sperm by microscopy, but could not detect MpBELL4-mScarlet protein expression in any of the seven lines. We nevertheless crossed some of the obtained lines with wild-type females (expecting undetectable but functional level of expression), but no complementation was observed.

As an alternative but less informative approach, we tried to demonstrate physical interaction between MpKNOX1 and MpBELL4 by in-vitro pull-down assay using recombinant proteins expressed in *E. coli*. While we could obtain recombinant MpKNOX1 His-tag fusion proteins, we could not obtain full-length MpBELL4 proteins for unknown reason. We thus separated MpBELL4 into three parts (Nter, Middle, and Cter) and expressed each as a GST-tagged protein with success. We tried pull-down experiments using a Glutathione-Sepharose resin, but we experienced a problem of His-MpKNOX1 proteins directly binding to the Sepharose resin. Consequently, we are currently unable to demonstrate interaction between MpKNOX1 and MpBELLs and its role in subcellular protein localization. We could continue our effort of expressing MpBELL-FP fusion proteins by other means, such as inserting an FP-coding sequence into the endogenous Mp*BELL* locus (producing knock-in lines), which is reported to be possible but still practically challenging. For protein interaction, the most promising approaches would be a BiFC assay, but it would be a simple repetition of the work done by Dierschke et al., With these persisting difficulties, we would now like to revise the corresponding part of discussion, admitting the weakness of our data and citing the work by Dierschke et al., to support.

2) Both maternal and paternal BELL3,4 are needed for fusion of pronuclei, but a considerable number of zygotes still went through successful fertilization in the absence of maternal/paternal BELL (78% and 63% respectively). This suggests that these "successful zygotes" use their 'uniparental' BELL to achieve karyogamy (or that other BELL proteins take over?). Therefore, it is expected that in the cross of MpKNOX-GFP female x Mpbell3,4 male (which retain GFP-KNOX in the cytosol) a large proportion of these zygotes would also achieve fertilization and therefore exhibit nuclear-localised KNOX (and BELL). However, it appears that zygotes from this cross do not have nuclearly-localized KNOX (and the authors do not mention how many replicate zygotes were used, an important piece of information that needs to be added). This is quite surprising. One would expect that more than half of the GFP lines would have a GFP signal from KNOX (and from BELL as well, if GFP-BELL was looked at) in the pronuclei. If none of the zygotes arising from this cross have GFP signal on the pronuclei, it suggests that successful fertilization can be achieved even in the absence of nuclear localization of KNOX? Please clarify.

We thank the reviewers for pointing out this critical issue. We admit that our phenotypic analysis was insufficient to quantitatively assess the contribution of paternal MpBELL3/4 on nuclear localization of MpKNOX1 and their relationship with zygote development. In response to this comment, we observed 26-48 zygotes derived from *MpKNOX1-GFP* eggs fertilized with either wild-type or Mp*bell3/4* sperm. The observation revealed that more than 80 % of zygotes produced with wild-type sperm exhibited nuclear-localized MpKNOX1-GFP at 12 HAF, whereas the ratio was less than a half for zygotes produced with Mp*BELL3/4* sperm (Figure 6G). The proportions of zygotes without nuclear-enriched MpKNOX1-GFP correlated well with the proportion of arrested zygotes (Figure 5M). Thus, our new data support the role of nuclear-localized MpKNOX1 in zygote activation. We added the corresponding discussion to the revised manuscript.

3. RNA-seq results of the Mprkd mutant (Figure 1) were not disclosed other than expression of MpKNOX1. Suggest including a table for the genes that were significantly more expressed in the wild type than in the Mprkd mutant archegonia.

A list of the genes significantly more expressed in wild-type than in Mp*rkd* archegonia was submitted as an Excel file (Table S1) in the original submission. We added another spreadsheet listing genes significantly more expressed in Mp*rkd* than in wild-type archegonia, and mentioned it in the main text.

4. MpKNOX1 is a single gene in the highly conserved KNOX1 family, predicting severe defects in the Mpknox1 mutant. As acknowledged by the authors, two independent Mpknox1-1ge and Mpknox1-2ge mutants showed incomplete defects and produced mature spores, indicating successful completion of sporophyte development. Results in Figure 4 need to be improved by providing quantitative details of zygote development in more than one knockout line, including fusion of pro-nuclei, # of capsules/sporangia, the germination rate of tetraspores,.

We now performed quantitative analyses of nuclear dynamics and embryo development of zygotes derived from wild-type and Mp*knox1* (three independent lines) eggs, each fertilized with wild-type sperm (Figure 4I-4K). We also quantified the number of sporangia per archegoniophore in wild-type and two independent Mp*knox1* female lines, each fertilized with wild-type males (Figure 4 Figure suppl. 4G). The new data revealed comparable defects in pronuclear fusion and sporangia formation in more than one Mp*knox1* alleles, supporting our postulation that MpKNOX1 is an important regulator of karyogamy and subsequent sporophyte development. While analysis of spore germination rate would be also interesting, we here focused our analysis on the phenotype of sporophyte development, because loss of Mp*KNOX1* functions had a major impact on the step of karyogamy.

5. The pronuclear fusion defect of Mpknox1-ko is one of the most critical observations, leading to the authors' conclusion that KNOX1 is required for zygote activation. Nuclear fusion itself is, however, not necessarily a pre-requisite for nor dependent on zygote activation. It is, therefore, important to report the pronuclear fusion rate at time points after 5 DAF. Provide quantitative analysis of pronuclear fusion and report whether zygote development proceeds with or without nuclear fusion.

We appreciate this important comments and performed quantitative analysis of pronuclear fusion in zygotes derived from Mp*knox1* eggs fertilized with wild-type sperm. At 5 DAF, all zygotes examined were arrested at karyogamy (Figure 4G and 4J). At 7DAF, although a small fraction (<10 %) of zygotes completed cell division (Figure 4K), we never observed unfused pronuclei in these developing zygotes. These data support that karyogamy is a necessary step to activate zygotic cell division in *M. polymorpha*, and that MpKNOX1 contributes to this step.

6. On the other hand, the leaky phenotypes of Mpknox1 mutants might be due to MpKNOX1 expression from the male genome in the zygotes of wild type male X Mpknox1-ko female, analogous to the Mpbell3/4-ko case presented in Figure 5. Since Figure 4 – Sup3 reported the phenotype of Mpknox1-1ge males, it is possible to examine Mpknox1-ko male X Mpknox1-ko female? This is critical for testing the two scenarios presented by the authors: leaky expression of MpKNOX1 and genetic redundancy.

We thank the reviewers for this constructive suggestion. In response to this comment, we crossed Mp*knox1* females (three independent lines) with either wild-type or Mp*knox1* males (two independent lines). For all Mp*knox1* alleles tested, a small fraction of sporophytes still developed from homozygous Mp*knox1* zygotes (Figure 4 – Figure suppl. 6). Consistently, paternal Mp*knox1* did not affect the number of sporangia per archegoniophore 4 weeks after fertilization (Figure 4 – Figure suppl. 4G). After all, this analysis did not distinguish the two possibilities, but rather emphasized a major contribution of maternal Mp*KNOX1* to zygote development. While the residual embryos formed from homozygous Mp*knox1* zygotes appears to argue for genetic redundancy, the Mp*knox1-6^ge^* allele used by Dierschke et al., in the accompanying paper, exhibited a complete penetrance in zygote arrest. So far, we are unsure about the origin of this discrepancy. It may be attributable to different genetic backgrounds (Tak-1 and Tak-2 in our study vs. MEL in Dierschke et al.,), but it is more likely that our Mp*knox1* alleles still produce partially functional MpKNOX1 proteins despite the presence of premature stop codons preceding the conserved HD domain (please note that Mp*knox1-6^ge^* generated by Dierschke et al., harbors a large deletion and is likely a null allele). We added description of these data and discussion to the revised manuscript.

7. Mpbell3/4 double mutants display leaky phenotypes. Incomplete arrest in early-stage zygote development might suggest genetic redundancy. Since the Marchantia genome possesses five BELL genes, comments about potential redundancy due to the other BELL family genes should be included. For instance, changes in MpKNOX1 localization is presented in support for a critical role of MpBELL3/4. Figure 6D shows residual MpKNOX1GFP signals in two nuclei at 12 HAF, and exclusion or disappearance of the signals from the nuclei at 24 HAF is evident in Figures 6C and 6E, suggesting that KNOX1 movement might occur in Mpbell3/4. This movement could be explained by MpBELL3/4 produced from the female nucleus or by redundant BELL.

We appreciate this comment. As stated in our response to Comment 2 above, our new data revealed that about a half of zygotes harboring paternally derived Mp*bell*3/4 still exhibited nuclear-enriched MpKNOX1-GFP signals (Figure 6G). Consistently, our phenotypic analysis indicates that MpBELL3/4 from both parents contribute to zygote development (Figure 5M). Thus, it is likely that maternally produced MpBELL3/4 make a certain contribution to the nuclear localization of MpKNOX1. On the other hand, some zygotes with homozygous Mp*bell3/4* still produced embryos, suggesting the existence of redundantly acting genes. Consistently, transcriptome data indicates weak expression of Mp*BELL1* and Mp*BELL2* in antheridiophores (Figure 5 —figure supplement 1A), and preferential expression Mp*BELL5* in archegonia (Figure 5 —figure supplement 1B). In response to the reviewers' suggestion, we included this discussion in the revised manuscript.

8. Zygote cell division at 5 DAF and nuclear translocation of MpKNOX1 in 12 HAF zygotes were reported as the primary defects of the Mpbell3/4 double mutants. Provide a quantitative assessment of MpKNOX1 nuclear localization between the BELL mutants and wild type. Also, provide details on their other phenotypes, such as whether the resulting zygotes develop into a partial embryo or produce spores. Such quantitative assessment is necessary for a complete evaluation of the MpBELL3/4 functions.

As stated in our response to Comment 2 above, we performed quantitative analysis of MpKNOX1-GFP localization (Figure 6G). The data indicates that about a half of the zygotes harboring paternally derived Mp*BELL3/4* did not exhibit nuclear-enriched GFP signals, and this ratio largely corresponded to the fraction of arrested zygotes at 5 DAG (Figure 5M). We also quantified the ratio of arrested zygotes at 7 DAF, as well as counted the number of sporangia per archegoniophore 4 weeks after fertilization (Figure 5 – Figure suppl. 3G and 3H). We found that about 70 % of zygotes harboring homozygous Mp*bell3/4* alleles were arrested at karyogamy. Consistently the number of sporangia produced from homozygous Mp*bell3/4* was about one third of those produced from wild-type zygotes. These data indicate that MpBELL3/4 regulate sporophyte development mainly at the step of karyogamy by promoting nuclear translocation of MpKNOX1 (and presumably MpBELL3/4 as well). We added description of these data and corresponding discussion to the revised manuscript.

9. While two Mpbell3/4 mutant lines were reported (Figure 4 – sup 2), Figures 5 and 6 include the results from only single Mpbell3/4 in the absence of genetic complementation. Provide the phenotypes of both lines for proper assessment of the Mpbell3/4 double knockouts.

We revised Figure 5 and 6 to include data obtained from two independent Mp*bell3/4* mutant alleles.